# The autophagy receptor p62/SQST-1 promotes proteostasis and longevity in *C. elegans* by inducing autophagy

Caroline Kumsta[1]*, Jessica T. Chang[1], Reina Lee[1], Ee Phie Tan[1], Yongzhi Yang[1], Rute Loureiro[2], Elizabeth H. Choy[1], Shaun H.Y. Lim [1], Isabel Saez[2], Alexander Springhorn[2], Thorsten Hoppe [2], David Vilchez [2] & Malene Hansen[1]*

Autophagy can degrade cargos with the help of selective autophagy receptors such as p62/SQSTM1, which facilitates the degradation of ubiquitinated cargo. While the process of autophagy has been linked to aging, the impact of selective autophagy in lifespan regulation remains unclear. We have recently shown in *Caenorhabditis elegans* that transcript levels of *sqst-1/p62* increase upon a hormetic heat shock, suggesting a role of SQST-1/p62 in stress response and aging. Here, we find that *sqst-1/p62* is required for hormetic benefits of heat shock, including longevity, improved neuronal proteostasis, and autophagy induction. Furthermore, overexpression of SQST-1/p62 is sufficient to induce autophagy in distinct tissues, extend lifespan, and improve the fitness of mutants with defects in proteostasis in an autophagy-dependent manner. Collectively, these findings illustrate that increased expression of a selective autophagy receptor is sufficient to induce autophagy, enhance proteostasis and extend longevity, and demonstrate an important role for *sqst-1/p62* in proteotoxic stress responses.

[1] Sanford Burnham Prebys Medical Discovery Institute, Development, Aging and Regeneration Program, 10901 North Torrey Pines Road, La Jolla, CA 92037, USA. [2] Institute for Genetics and Cologne Excellence Cluster for Cellular Stress Responses in Aging-Associated Diseases (CECAD), University of Cologne, Joseph Stelzmann Strasse 26, 50931 Cologne, Germany. *email: ckumsta@sbpdiscovery.org; mhansen@sbpdiscovery.org

Macroautophagy (hereafter called autophagy) facilitates degradation and recycling of cytosolic components, referred to as cargo, in response to nutrient deprivation or other stressors. Autophagy is initiated by the nucleation of a double membrane, which forms the phagophore. As the phagophore expands, it begins to sequester cytosolic cargo into the growing vesicle. Upon completion, the autophagosome or amphisome (formed by fusion with vesicles from the endolysosomal compartments) then fuse with acidic lysosomes, resulting in the degradation of the sequestered content by hydrolases. Autophagy is essential for survival, development, and organismal homeostasis[1]. It occurs at low levels under basal conditions, whereas developmental stimuli or cellular stress, including starvation and heat shock (HS), can induce autophagy[1,2]. Furthermore, autophagy can protect against pathologies, including neurodegeneration and aging[3]. The regulation of autophagy with age is incompletely understood, but several lines of evidence suggest that autophagy declines with age[3]. Conversely, autophagy genes are essential for lifespan extension in distinct longevity paradigms in *S. cerevisiae*, *C. elegans*, and *Drosophila* (for review see ref. [3]). While these observations demonstrate a link between autophagy and aging, it remains unclear how the autophagy process affects longevity and healthspan.

Autophagy was originally described as a 'bulk' turnover process, in which cytosolic components are indiscriminately recycled to provide amino acids and other building blocks during nutrient deprivation and cellular stress[4]. Emerging evidence indicates that selective types of autophagy degrade specific and possibly damaged cytosolic components in a tightly regulated manner. During selective autophagy, autophagosomes recruit specific types of cargo, including mitochondria and protein aggregates, through the action of autophagy receptors that connect the autophagosome to the cargo[4]. The binding of the autophagy receptors is facilitated by Atg8/LC3/GABARAP family proteins, which are proteolytically cleaved and conjugated with phosphatidylethanolamine and incorporated into the forming autophagosomal membranes[4]. The selective autophagic degradation of ubiquitinated protein aggregates, termed aggrephagy, can be facilitated by autophagy receptor p62/SQSTM1 (hereafter referred to as p62). p62 has several conserved domains that are important for its function in autophagy. The C-terminal ubiquitin-associated (UBA) domain enables p62 to bind to ubiquitinated proteins[4]. The LC3-interacting region (LIR domain) allows binding of p62 to Atg8/LC3/GABARAP family members, and thus facilitates degradation of ubiquitinated protein cargo[5,6]. p62-mediated aggrephagy also requires the Phox-and-Bem1 (PB1) domain of p62, which allows its self-oligomerization and strengthens binding to the Atg8/LC3-containing autophagic membranes[7].

The degradation of ubiquitinated proteins can occur via autophagy as well as the ubiquitin-proteasome system (UPS), and p62 has been implicated in both processes; i.e., as a selective autophagy receptor, and through the delivery of ubiquitinated proteins for degradation to the proteasome[8–12]. p62 is constantly being degraded by autophagy[5], and inhibition of autophagy leads to the accumulation of p62 aggregates in mice[7], *Drosophila*[13], and *C. elegans*[14]. These ubiquitin-positive aggregates are significantly reduced in autophagy-deficient mice and flies when *p62* is deleted[13,15], indicating a role for p62 in aggregate formation. Furthermore, p62 has been found associated with ubiquitin-positive aggregates observed in neurodegenerative-disease models, such as Huntington's and Alzheimer's disease[13]. The biological relevance of aggregate formation is still under debate and may depend on the cellular context and the size of the aggregates. To this end, it has been suggested that larger aggregates are less toxic to the cell than smaller aggregates dispersed throughout the cell[16–18]. While p62 overexpression can enhance protein aggregation, overexpression of p62 has protective effects on cell survival[5] and p62 has an important role in preventing neurodegenerative diseases[10,12], although the exact protective mechanisms are still unclear.

Consistent with a key role in age-related disease, mice deficient in *p62* have reduced lifespan, increased oxidative stress[19], synaptic deficiencies, and memory impairment[20], albeit neither significant neurodegeneration nor aggregation formation are observed in neurons of these mice[15]. Older $p62^{-/-}$ mice, however, accumulate insoluble ubiquitinated proteins in the brain[21]. Similar to mice, *Drosophila* expressing mutant Ref(2)P/p62, which lacks either the PB1 or the UBA domain, are short-lived, have impaired movement, and mitochondrial defects[22]. In contrast, in *C. elegans*, mutations in the *p62* ortholog *sqst-1* cause no lifespan defects[23,24], indicating a limited requirement for *sqst-1/p62* (hereafter called *sqst-1*) under basal conditions in the nematode.

The expression levels of *p62* have been shown to decline with age in mice[19], and reduced expression of *p62* correlates with age-related neurodegenerative diseases in humans[25,26]. Notably, we recently reported that *sqst-1* mRNA levels are markedly increased upon heat stress in *C. elegans*[2], prompting the hypothesis that SQST-1 may play a role in the heat-shock response, in which heat-shock proteins and molecular chaperones are rapidly and transiently induced to ameliorate the deleterious effects of heat stress[27]. We further reported that exposure of *C. elegans* to a hormetic HS, i.e., a short stimulus with increased temperature that would be lethal at prolonged exposure, not only increases stress resistance and longevity, but also improves proteostasis in an autophagy-dependent manner[2]. Since emerging evidence suggests that the degradation of specific cargos by selective autophagy is important for maintaining health[3], we investigated the role of SQST-1 in hormetic HS, lifespan, and proteostasis. Here we demonstrate that *sqst-1* is required for autophagy induction as well as organismal benefits conferred by a hormetic HS. Furthermore, we show that overexpression of full-length SQST-1, but not truncated SQST-1 lacking the UBA domain, is sufficient to increase longevity in *C. elegans*. SQST-1 overexpression leads to tissue-specific induction of autophagy and decreases aggregation of expanded PolyQ proteins as well as temperature-sensitive protein-folding mutants in an autophagy-dependent manner. These observations illustrate that overexpression of a selective autophagy receptor is sufficient to induce autophagy and enhance longevity and proteostasis. As p62 plays an important role in many age-related diseases, our findings highlight potential therapeutic opportunities in inducing *p62*-mediated selective autophagy.

## Results

**sqst-1 regulates basal autophagy in a tissue-specific manner.** A hormetic heat shock (HS) early in life of *C. elegans* induces autophagy and increases the transcription of autophagy-related genes, including *sqst-1*[2]. HS causes proteotoxic stress, which is accompanied by protein unfolding and can ultimately lead to protein aggregation[28,29]. Protein aggregates and ubiquitinated protein conjugates accumulate upon a HS of 1 h at 36 °C in *C. elegans* (Supplementary Fig. 1a, b), as observed in human melanoma cells[30]. Since ubiquitinated proteins can be degraded by selective autophagy via the autophagy receptor p62[5], and a hormetic HS induces an increase of autophagosomes in *C. elegans*[2], we asked if *sqst-1* could play a role in the regulation of autophagy under heat-stress conditions. To this end, we first examined whether reduction of *sqst-1* function would lead to autophagy defects under basal conditions. While SQST-1 protein expression appears limited to the pharynx and nerve-ring

neurons under basal conditions, *sqst-1* is transcriptionally expressed in nearly all tissues in *C. elegans*[14,31,32]. Therefore, we monitored autophagy in nerve-ring neurons, intestine, body-wall muscle, and pharynx of *sqst-1(ok2892)* mutants, which expresses a truncated loss-of-function SQST-1 protein[24,33], and in a CRISPR-engineered, full-length deletion of *sqst-1 (sqst-1(syb764))* expressing GFP-tagged LGG-1/Atg8[34–36]. This autophagy reporter visualizes autophagic structures (i.e., pre-autophagosomal structures, autophagosomes, and amphisomes) as fluorescent punctae.

Both *sqst-1* mutants showed a decrease in the number of GFP::LGG-1/Atg8-positive punctae in nerve-ring neurons (Fig. 1a and Supplementary Fig. 2a, b), whereas the number of GFP::LGG-1/Atg8-positive punctae was unchanged in the intestine, body-wall muscle, or pharynx (Fig. 1b–d and Supplementary Fig. 2c–h) [For hypodermal seam cells, see Supplementary Fig. 2i, j]. Concomitant analysis of a lipidation-deficient GFP::LGG-1(G116A) mutant[37] confirmed that LGG-1/Atg8-positive structures were likely to represent autophagosomes in these four major tissues (Supplementary Fig. 2). Since the two *sqst-1* mutants displayed the same autophagy phenotypes and were not compromised in the healthspan measures we investigated, i.e., movement, pharyngeal pumping, or progeny production (Supplementary Fig. 3), we used the *sqst-1(ok2892)* mutants in our subsequent analyses, unless stated otherwise.

Monitoring the number of GFP::LGG-1/Atg8-positive punctae does not conclusively inform on autophagic activity[38]. Therefore, we also performed autophagy-flux assays in *sqst-1(ok2892)* mutants by injecting the autophagy inhibitor bafilomycin A1 (BafA), which blocks autophagosomal turnover[39,40]. A change in the number of GFP::LGG-1/Atg8-positive punctae upon BafA addition indicates that autophagy is active, whereas no change indicates that the cell/tissue is experiencing a block in autophagy. Notably, BafA treatment did not increase the number of GFP::LGG-1/Atg8-positive punctae in the nerve-ring neurons of *sqst-1(ok2892)* mutants, which had very low numbers of autophagosomes under basal conditions (Fig. 1a), Taken together, these findings indicate that *sqst-1* is required for autophagosome formation in neurons, as observed in HeLa cells[7], rather than increasing the rate of autophagosomal turnover. In contrast, the intestine (Fig. 1b), body-wall muscles (Fig. 1c), as well as the pharynx (Fig. 1d) showed a clear increase in the number of GFP::LGG-1/Atg8-positive punctae after BafA injection, consistent with autophagic turnover under basal conditions in non-neuronal tissues. In sum, these results indicate that *sqst-1(ok2892)* animals display a tissue-specific block in autophagy in nerve-ring neurons.

**sqst-1 is required for autophagy induction upon heat shock**. To determine whether *sqst-1* could play a role in the regulation of autophagy in response to heat stress, we next asked whether a hormetic HS would induce autophagy in different tissues of *sqst-1 (ok2892)* mutants. In contrast to WT animals[2], exposure of *sqst-1 (ok2892)* mutants to a hormetic HS failed to induce autophagosome accumulation in nerve-ring neurons (Fig. 1e), intestine (Fig. 1f), body-wall muscle (Fig. 1g), and pharynx (Fig. 1h), indicating that *sqst-1* is broadly required for autophagy induction in many tissues upon heat stress. Since we previously showed that a hormetic HS increases the mRNA levels of autophagy-related genes[2], we performed quantitative RT-PCR analyses in *sqst-1 (ok2892)* mutants, and found that autophagy-related genes were increased in *sqst-1(ok2892)* mutants after HS similarly to WT animals (Supplementary Fig. 4a). *sqst-1(ok2892)* mutants similarly induced the heat-shock responsive gene *hsp-70* compared

with WT animals (Supplementary Fig. 4b). Since *sqst-1(ok2892)* mutants could induce heat-shock responsive- and autophagy genes following a hormetic HS, *sqst-1* may function downstream of the transcriptional responses induced by HS. Altogether, our data suggest that HS-induced autophagy is *sqst-1*-dependent in the entire organism, whereas *sqst-1* is required for basal autophagy specifically in neurons.

**sqst-1 is required for the benefits of a hormetic heat shock**. In addition to inducing autophagy, a hormetic HS early in life of *C. elegans* increases longevity, improves stress resistance, and is sufficient to prevent aggregation of intestinal and neuronal expanded PolyQ proteins[2]. Since we found *sqst-1* to be required for autophagy induction after a hormetic HS, we asked whether *sqst-1* could help mediate the benefits of a hormetic HS. Notably, *sqst-1(ok2892)* mutants did not benefit from a hormetic HS on day 1 of adulthood as measured by thermo-resistance (Fig. 2a and Supplementary Table 1), or longevity (Fig. 2b and Supplementary Table 2), demonstrating that *sqst-1* is required for the positive effect of hormetic HS on thermo-resistance and longevity.

In contrast, *sqst-1* was not required for the long lifespan of other conserved longevity paradigms that have been linked to autophagy[3], including *daf-2*/insulin/IGF-1 receptor mutants, germline-less *glp-1* mutants, animals with reduced mTOR levels (*let-363* RNAi), and animals with reduced levels of components of the mitochondrial electron transport chain (*cyc-1* RNAi) (Supplementary Data 1). This indicates that *sqst-1*, unlike core autophagy genes[3], is not broadly implicated in lifespan extension in *C. elegans*, but instead is required under specific, potentially proteotoxic conditions, such as a hormetic HS.

Finally, to determine whether selective autophagy mediated by *sqst-1* could be involved in the removal of protein aggregates, we used *C. elegans* expressing tissue-specific YFP-tagged PolyQ proteins[41–43]. In *C. elegans*, PolyQ proteins accumulate with age[41], and have been shown to be ubiquitinated[44]. In contrast to the increased PolyQ aggregation observed upon reduction of core autophagy genes[2,45], PolyQ aggregation was unchanged under basal conditions in neurons of *sqst-1(ok2892)* mutants (Fig. 2c). This result suggests that *sqst-1*-dependent autophagy is not required for basal degradation of neuronal PolyQ aggregates. In contrast to WT animals in which a hormetic HS reduces aggregation[2], *sqst-1(ok2892)* animals expressing neuronal PolyQ proteins showed an increased aggregate load in neurons following a hormetic HS (Fig. 2c). Moreover, fluorescence recovery after photobleaching (FRAP) experiments revealed that HS increased the mobility of the neuronal PolyQ aggregates in WT, but not in *sqst-1(ok2892)* mutant animals (Fig. 2d, e), while having only limited effects on diffuse neuronal PolyQ proteins (Supplementary Fig. 5). Consistently, a hormetic HS increased the soluble portion and decreased the insoluble portion of PolyQ proteins in WT, but not in *sqst-1(ok2892)* mutant animals (Supplementary Fig. 6). Similar to aggregate load (Fig. 2c), the baseline values for WT and *sqst-1(ok2892)* animals were not significantly different in FRAP (Fig. 2d, e) or in differential detergent extraction experiments (Supplementary Fig. 6). Furthermore, *sqst-1(ok2892)* mutants expressing neuronal PolyQ proteins were not significantly longer lived following a hormetic HS (Supplementary Table 2). Collectively, these assays show that *sqst-1* is required for the benefits of a hormetic HS on neuronal proteostasis. In contrast, *sqst-1* was only partially required for intestinal proteostasis in terms of PolyQ aggregation, while muscle-specific PolyQ aggregation was unaffected by a hormetic HS or *sqst-1(ok2892)* mutation (Supplementary Fig. 7 and Supplementary Table 2, i.e., aggregate number, FRAP, and lifespan). Taken together, these data suggest that *sqst-1* is required when

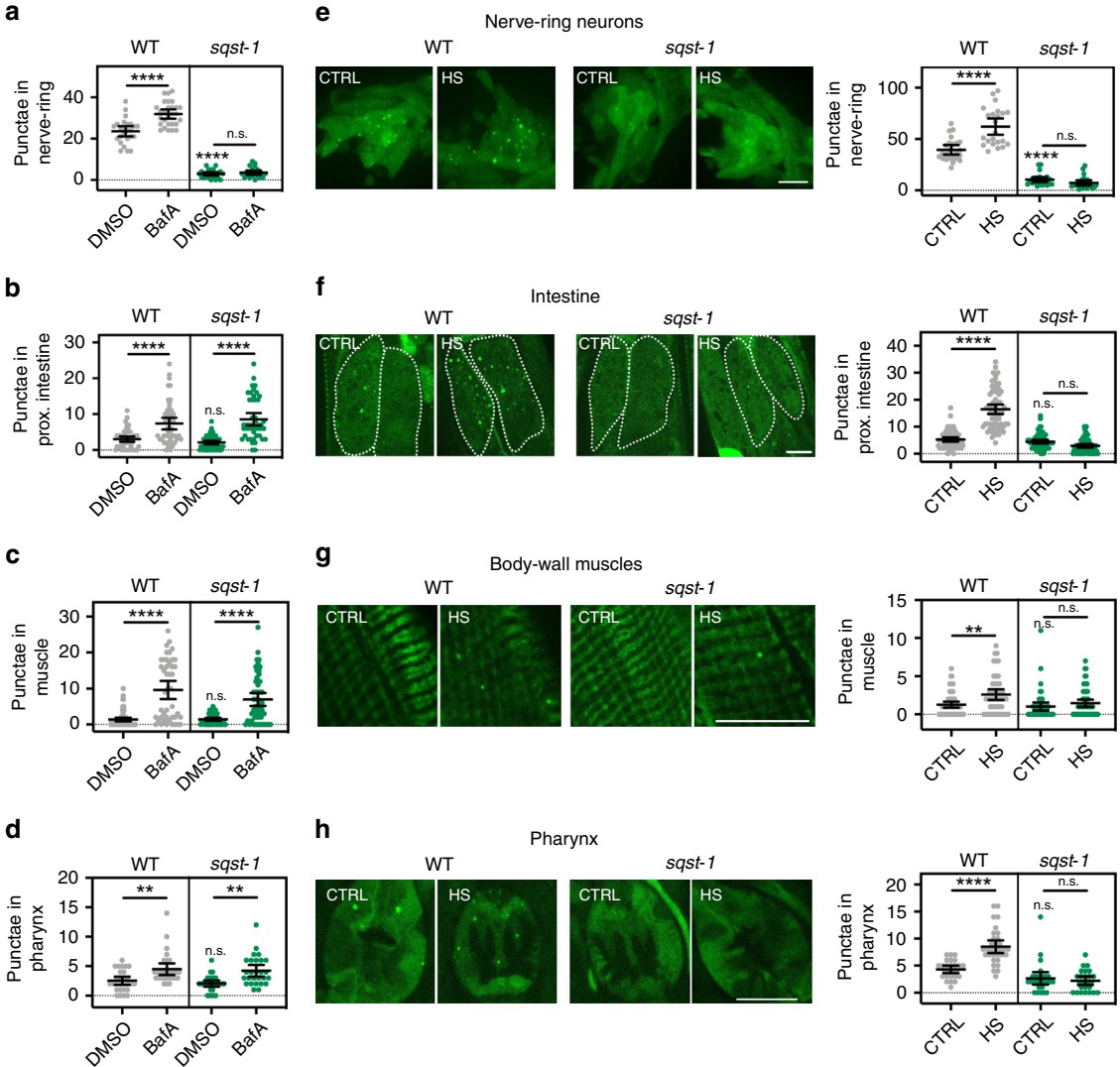

**Fig. 1** *sqst-1* is required for autophagy induction after hormetic heat shock (HS). **a–d** Autophagy flux was measured at 20 °C in WT and *sqst-1(ok2892)* (*sqst-1*) animals on day 1 of adulthood expressing *rgef-1p::gfp::lgg-1* (**a**) or *lgg-1p::gfp::lgg-1* (**b–d**). WT and *sqst-1* animals were injected with vehicle (DMSO) or bafilomycin A1 (BafA) to block autophagy at the lysosomal acidification step. GFP::LGG-1/Atg8-positive punctae were quantified from three independent experiments in **a** nerve-ring neurons (WT-DMSO, N = 25; WT-BafA, N = 28; *sqst-1*-DMSO, N = 25, *sqst-1*-BafA, N = 24 animals), **b** proximal intestinal cells (WT-DMSO, N = 49; WT-BafA, N = 50; *sqst-1*-DMSO, N = 55, *sqst-1*-BafA, N = 46 cells), **c** body-wall muscle (WT-DMSO, N = 71; WT-BafA, N = 44; *sqst-1*-DMSO, N = 48, *sqst-1*-BafA, N = 53 animals), and **d** and the terminal pharyngeal bulb (WT-DMSO, N = 28; WT-BafA, N = 29; *sqst-1*-DMSO, N = 28, *sqst-1*-BafA, N = 27 animals), Error bars indicate 95% CI. ns: $P > 0.05$, $**P < 0.01$, $****P < 0.0001$, by two-way ANOVA with Tukey's multiple comparisons test. **e–h** GFP::LGG-1/Atg8-positive punctae were counted in WT and *sqst-1(ok2892)* (*sqst-1*) animals on day 1 of adulthood expressing *rgef-1p::gfp::lgg-1* (**e**) or *lgg-1p::gfp::lgg-1* (**f–h**). Animals were maintained under control conditions 15 °C (CTRL) or subjected to HS for 1 h at 36 °C (HS). Animals were imaged at time of highest induction in autophagy as previously characterized[2]. GFP::LGG-1/Atg8 punctae were quantified from three independent experiments in **e** nerve-ring neurons (WT-CTRL, N = 24; WT-HS, N = 23; *sqst-1*-CTRL, N = 23, *sqst-1*-HS, N = 24 animals) with 2 h of recovery after HS, **f** proximal intestinal cells (WT-CTRL, N = 67; WT-HS, N = 70; *sqst-1*-CTRL, N = 66, *sqst-1*-HS, N = 71 cells) with 4 h of recovery after HS, **g** body-wall muscle (WT-CTRL, N = 60; WT-HS, N = 48; *sqst-1*-CTRL, N = 50, *sqst-1*-HS, N = 60 animals) (no recovery), and **h** terminal pharyngeal bulb (WT-CTRL, N = 26; WT-HS, N = 32; *sqst-1*-CTRL, N = 26, *sqst-1*-HS, N = 25 animals) with 4 h of recovery after HS. Error bars indicate 95% CI. ns: $P > 0.05$, $**P < 0.01$, $****P < 0.0001$, by two-way ANOVA with Tukey's multiple comparisons test. Scale bar: 10 μm. Source data are provided in the Source Data file.

autophagy is induced by a hormetic HS and plays a key role in improving proteostasis particularly in neurons of PolyQ-expressing animals.

**Overexpression of SQST-1 is sufficient to extend lifespan**. We previously showed that a hormetic HS leads to an increase in the transcription of *sqst-1*[2], and observed that *sqst-1* mRNA levels remained upregulated for several days after a HS (Supplementary Fig. 4c). Given the requirement of *sqst-1* for HS-induced

autophagy (Fig. 1) and the organismal benefits of a hormetic HS (Fig. 2), we next tested whether increasing *sqst-1* levels in *C. elegans* was sufficient to induce organismal benefits, similar to a hormetic HS. To this end, we overexpressed *sqst-1* fused to *gfp* under the control of its endogenous promoter and found SQST-1::GFP to be expressed mainly in punctate structures in head neurons and in the pharynx throughout adulthood (Supplementary Fig. 8a). SQST-1::GFP accumulated upon autophagy block (Supplementary Fig. 8b), as observed for untagged SQST-1[31], indicating that the SQST-1::GFP protein is

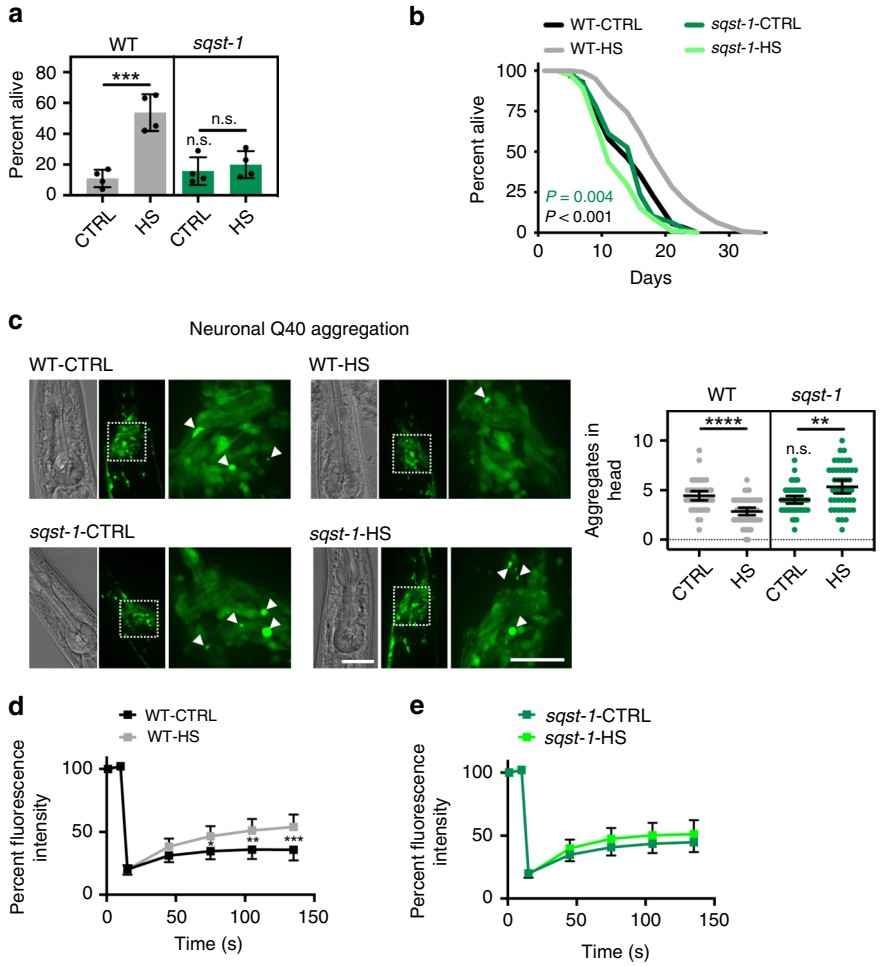

**Fig. 2** *sqst-1* is required for the organismal benefits of a hormetic heat shock (HS). **a** Survival of WT (N2) and *sqst-1(ok2892)* (*sqst-1*) animals subjected to HS on day 1 of adulthood, and then incubated for 7 h at 36 °C on day 3 of adulthood (*n* = 4 plates). Error bars indicate SD, ns: *P* > 0.8, ***P* < 0.001 by two-way ANOVA with Tukey's multiple comparisons test. See Supplementary Table 1 for experimental details and additional repeats. **b** Lifespan analysis of WT and *sqst-1(ok2892)* (*sqst-1*) animals at 20 °C subjected to 1h HS at 36 °C on day 1 of adulthood. WT-CTRL animals (*N* = 126) compared with WT-HS animals (*N* = 110): *P* < 0.0001; *sqst-1*-CTRL animals (*N* = 128) (*P* = 0.3 to WT) compared with *sqst-1*-HS animals (*N* = 115): *P* = 0.004, by log-rank test. See Supplementary Table 2 for experimental details and additional repeats. **c** Neuronal PolyQ (*rgef-1p::Q40::yfp*) aggregates detected on day 4 of adulthood in WT and *sqst-1(ok2892)* (*sqst-1*) animals maintained under control conditions (20 °C, CTRL) or subjected to HS (1 h at 36 °C) on day 1 of adulthood (HS). Scale bar 20 μm. The number of neuronal PolyQ aggregates were quantified from three independent experiments in WT-CTRL, *N* = 43; WT-HS, *N* = 50; *sqst-1*-CTRL, *N* = 48, *sqst-1*-HS, *N* = 45 animals. Error bars indicate 95% CI. ns: *P* > 0.6, ***P* < 0.01, *****P* < 0.0001, by two-way ANOVA with Tukey's multiple comparisons test. Scale bar 20 and 10 μm for close-up. **d**, **e** FRAP measurements of neuronal PolyQ (*rgef-1p::Q40::yfp*) aggregates on day 8 of adulthood in WT (**e**) and *sqst-1(ok2892)* (*sqst-1*) animals (**f**) maintained under control conditions (20 °C, CTRL) or subjected to HS (1 h at 36 °C) on day 1 of adulthood (HS). The fluorescence signal was bleached after 20 s and fluorescence recovery was measured for 2 min. The average % of fluorescence intensity was quantified from three independent experiments in WT-CTRL, *N* = 32; WT-HS, *N* = 30; *sqst-1*-CTRL, *N* = 32, *sqst-1*-HS, *N* = 31 animals. Error bars indicate 95% CI. ***P* < 0.05, ***P* < 0.01, ****P* < 0.001, by two-way ANOVA with Sidak's multiple comparisons test. *P* > 0.05 for all timepoints when comparing WT-CTRL and *sqst-1*-CTRL by two-way ANOVA with Sidak's multiple comparisons test. Source data are provided in the Source Data file.

functional as it can serve as a substrate for autophagy. Animals overexpressing SQST-1 (GFP-tagged or untagged), either from extrachromosomal arrays or from integrated loci, displayed an average lifespan extension of ~30% (Fig. 3a and Supplementary Data 2). Integrated SQST-1::GFP-expressing animals had a ~30–100-fold increase in SQST-1 levels, whereas animals over-expressing untagged SQST-1 from extrachromosomal arrays in the *sqst-1(syb764)* mutant background had a ~2–7-fold increase in SQST-1 levels (Supplementary Fig. 8c, d), highlighting that a range of SQST-1 overexpression resulted in lifespan extension. Importantly, in the *sqst-1(syb764)* mutant, overexpression of an untagged, truncated SQST-1 protein without the UBA domain

(SQST-1ΔUBA), which enables p62 to bind ubiquitinated pro-teins[4], did not extend lifespan (Fig. 3b, Supplementary Data 2; Supplementary Fig. 8d). These observations indicate that the interaction of SQST-1 with ubiquitinated proteins upon SQST-1 overexpression is required for the lifespan-extending effects of SQST-1 overexpression.

Given that *sqst-1* was required for autophagosome formation in neurons (Fig. 1a, e), and because the SQST-1::GFP expression was primarily observed in neurons surrounding the pharynx (Sup-plementary Fig. 8a), we overexpressed SQST-1::GFP under the control of a pan-neuronal promoter (*rgef-1p*). Transgenic animals overexpressing neuronal SQST-1::GFP (>100-fold increase in

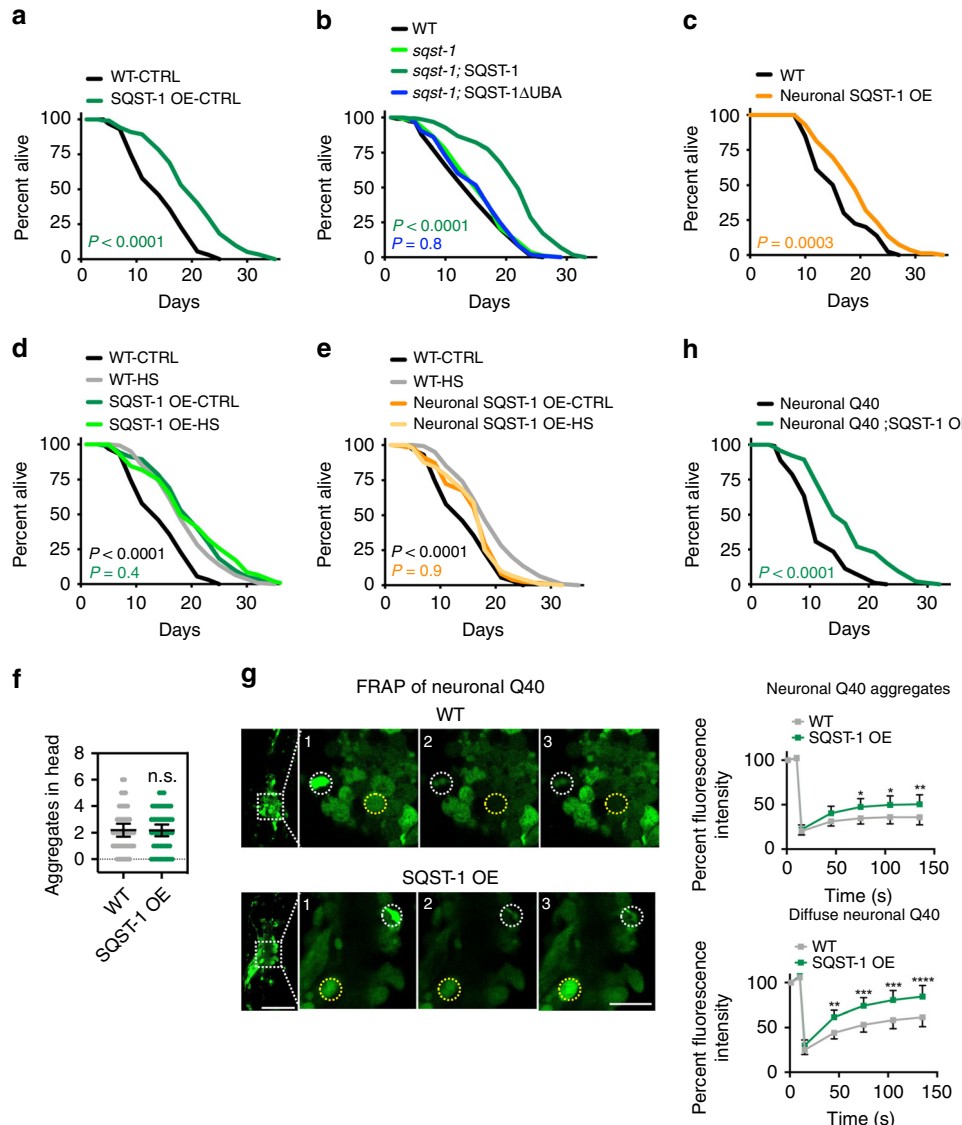

**Fig. 3** SQST-1 overexpression extends *C. elegans* lifespan and improves proteostasis. **a** Lifespan analysis of WT (N2) ($N = 58$ animals) and SQST-1 OE (*sqst-1p::sqst-1::gfp* (1)) ($N = 88$ animals) animals at 20 °C: *P*-value by log-rank test. **b** Lifespan analysis of WT (N2) ($N = 92$ animals), *sqst-1(syb764)* ($N = 88$ animals), and *sqst-1(syb764)* animals expressing either *sqst-1p::sqst-1 + rol-6* (*sqst-1;* SQST-1) ($N = 124$ animals) or *sqst-1p::sqst-1ΔUBA + rol-6* (*sqst-1;* SQST-1ΔUBA) ($N = 107$ animals) at 20 °C. *P*-values by log-rank test. **c** Lifespan analysis of WT (N2) ($N = 85$ animals) and neuronal SQST-1 OE (*rgef-1p:: sqst-1::gfp* (1)) ($N = 85$ animals) animals at 20 °C. *P*-value by log-rank test. **d** Lifespan analysis of WT (N2) and SQST-1 OE (*sqst-1p::sqst-1::gfp* (1)) animals subjected to hormetic HS on day 1 of adulthood. WT-CTRL, $N = 126$; WT-HS, $N = 111$; SQST-1 OE-CTRL, $N = 116$, SQST-1 OE-HS, $N = 106$ animals at 20 °C. *P*-values by log-rank test. **e** Lifespan analysis of WT (N2) and neuronal SQST-1 OE (*rgef-1p::sqst-1::gfp* (1)) animals subjected to hormetic HS on day 1 of adulthood. WT-CTRL, $N = 126$; WT-HS, $N = 111$; Neuronal SQST-1 OE-CTRL, $N = 123$, SQST-1 OE-HS, $N = 120$ animals at 20 °C. *P*-values by log-rank test. **a–e** See Supplementary Data 2 and Table 3 for experimental repeats. **f** Quantification of neuronal PolyQ aggregates (*rgef-1p::Q40::yfp*) on day 5 of adulthood from three independent experiments in WT ($N = 31$ animals) and SQST-1 OE animals (*sqst-1p::sqst-1 + rol-6*) ($N = 30$ animals). Error bars indicate 95% CI. ns: $P > 0.05$ by *t*-test. **g** Fluorescence images and FRAP measurements of the aggregated (white dashed circles) and diffuse (yellow dashed circles) portion of neuronal PolyQ (*rgef-1p::Q40::yfp*) on day 8 of adulthood in WT and SQST-1-overexpressing animals *sqst-1p::sqst-1 + rol-6* (SQST-1 OE) at 20 °C. Fluorescence signal bleached after 10 s (timepoint 2) and fluorescence recovery was measured for 2 min. The average % of fluorescence intensity was quantified from three independent experiments in WT, $N = 32$; SQST-1 OE, $N = 31$. Image numbers correspond to the indicated timepoints. Error bars indicate 95% CI. All measurements are $P > 0.05$ by two-way ANOVA with Sidak's multiple comparisons test unless indicated otherwise *$P < 0.05$, **$P < 0.01$, ***$P < 0.001$, ****$P < 0.0001$. Outline of pharyngeal bulbs in image of head region. Scale bar: 20 and 10 μm for close-up. **h** Lifespan analysis of Neuronal Q40-expressing animals (*rgef-1p::Q40::yfp*) ($N = 39$ animals) and Neuronal Q40; SQST-1 OE (*rgef-1p::Q40::yfp; sqst-1p:: sqst-1 + rol-6*) ($N = 79$ animals) at 20 °C. *P*-value by log-rank test. See Supplementary Table 4 for experimental repeats. Source data are provided in the Source Data file.

integrated lines) displayed up to ~25% lifespan extension at 20 °C, albeit with some variability and most consistently when SQST-1 was expressed from extrachromosomal arrays (Fig. 3c, Supplementary Data 2 and Supplementary Table 3). Of note, our collection of SQST-1::GFP-overexpressing strains generally displayed healthspan parameters similar to WT animals (Supplementary Fig. 9). Collectively, these data show that overexpression of a selective autophagy receptor, SQST-1, is sufficient to extend *C. elegans* lifespan.

Since we found that *sqst-1* was required for the lifespan increase mediated by a hormetic HS (Fig. 2d), we tested whether subjecting SQST-1-overexpressing animals to a hormetic HS would be additive for lifespan. The lifespan of animals overexpressing SQST-1::GFP under the endogenous promoter or under the neuronal *rgef-1* promoter was unchanged after a hormetic HS (Fig. 3d, e and Supplementary Table 3). These results, together with the requirement of *sqst-1* for the benefits of the hormetic HS (Fig. 2), are consistent with hormetic HS and SQST-1 overexpression extending lifespan by common mechanisms.

**SQST-1 overexpression improves proteostasis**. Since *sqst-1* was required for hormetic HS to alleviate neuronal PolyQ aggregation (Fig. 2c), we next investigated whether SQST-1 overexpression could similarly alleviate neuronal PolyQ aggregation. While the number of neuronal PolyQ aggregates in SQST-1-overexpressing animals was unchanged (Fig. 3f), the mobility of both the aggregated as well as the diffuse PolyQ proteins was increased upon overexpression of SQST-1 (Fig. 3g). These data suggest that SQST-1 overexpression changes the properties of aggregation-prone PolyQ::YFP. Furthermore, SQST-1 overexpression was sufficient to extend the lifespan of animals expressing neuronal PolyQ proteins (Fig. 3h), as well as animals carrying temperature-sensitive (ts) missense mutations in genes causing protein misfolding and tissue-specific protein aggregation[46,47], i.e., dynamin GTPase (*dyn-1(ts);* neurons), Ras (*let-60(ts);* intestine), and myosin (*unc-54(ts);* muscle) (Supplementary Table 4). Taken together, SQST-1 overexpression was able to improve proteostasis and longevity in *C. elegans* with protein-folding defects.

p62 can mediate the degradation of aggregation-prone proteins[8–12]. Therefore, we next asked by which mechanisms SQST-1 overexpression mediated proteostasis in *C. elegans*. We found that both endogenous and neuronal overexpression of SQST-1::GFP reduced the levels of ubiquitinated proteins compared with WT animals (Supplementary Fig. 10a). Since the degradation of ubiquitinated proteins can occur via the ubiquitin-proteasome system (UPS) as well as autophagic degradation, we next tested whether SQST-1-overexpressing animals exhibited elevated proteasome activity using a fluorogenic peptide substrate specific for the chymotrypsin-like activity of the proteasome[48]. We found that *glp-1* mutants, previously reported to exhibit increased proteasomal activity[48], had an increase of 100% in their proteasome activity, whereas animals overexpressing SQST-1 under the endogenous promoter (SQST-1-overexpressing line 1 and 2) had either a ~10% marginal or ~30% significant increase, respectively; and animals overexpressing SQST-1 in the neurons alone had a modest ~15% increase (Supplementary Fig. 10b). Since the proteasomal activity of SQST-1-overexpressing animals was only modestly increased, we next examined whether an induction of autophagy could contribute to the increased organismal health and proteostasis upon SQST-1 overexpression. First, we investigated if autophagy was required for the lifespan extension observed in SQST-1-overexpressing animals. We fed SQST-1-overexpressing animals bacteria expressing dsRNA targeting distinct autophagy-related genes from adulthood on.

We found that RNAi targeting autophagy genes involved in different steps of the autophagy process, such as *lgg-1/ATG8* (Fig. 4a), which is part of the Atg8-conjugation system, or *cup-5* (Fig. 4b), whose loss impairs degradation of autolysosomal contents[49], abrogated the lifespan extension of SQST-1-overexpressing animals. A similar requirement was found for *vps-34* and *bec-1/Atg6*, which are involved in nucleation of autophagosomal membranes, *atg-18*, which is involved in phagophore formation, and *epg-5*, which is involved in lysosomal degradation[50] (Supplementary Data 3). Since multiple genes with functions in different steps of the autophagy process were similarly required for the lifespan extension conferred by overexpression of SQST-1, autophagy as a general process is likely required for the lifespan extension of SQST-1-overexpressing animals.

To explore whether autophagy was required for the effects of SQST-1-overexpression on proteostasis, we again used *dyn-1(ts),*

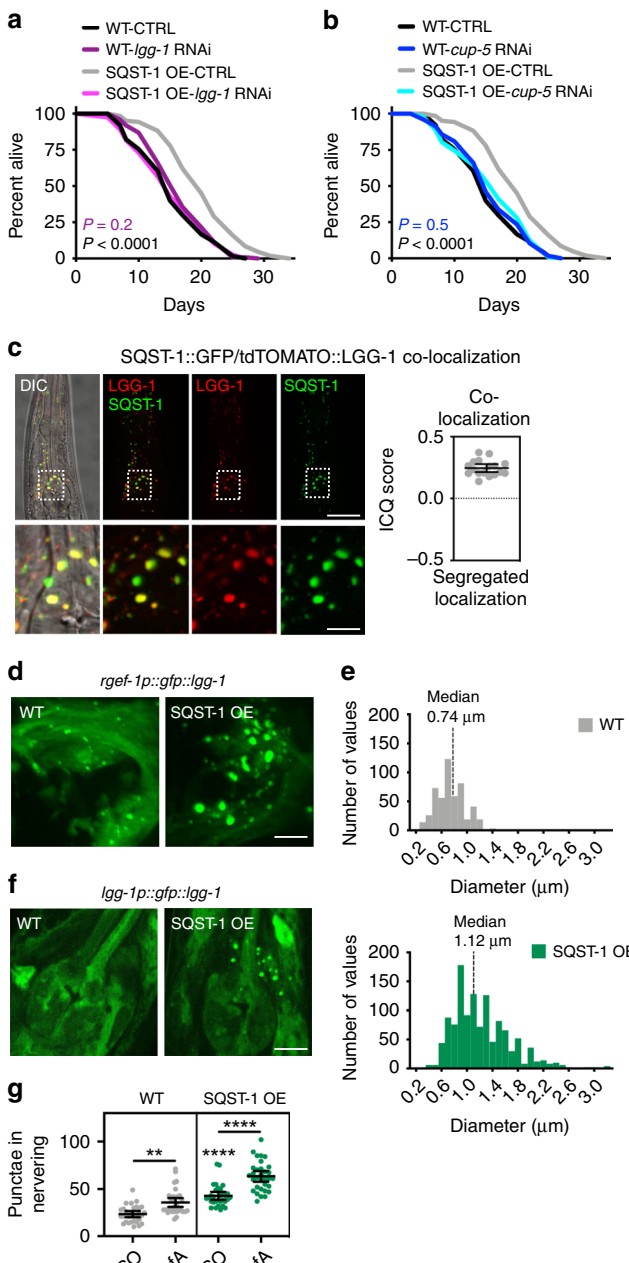

**c** SQST-1::GFP/tdTOMATO::LGG-1 co-localization

**d** *rgef-1p::gfp::lgg-1*

**e** Median 0.74 µm — WT

**f** *lgg-1p::gfp::lgg-1*

SQST-1 OE Median 1.12 µm

**g**

**Fig. 4** SQST-1 overexpression induces neuronal autophagy. **a**, **b** Lifespan analysis of WT and *sqst-1p::sqst-1::gfp* (SQST-1 OE) animals at 20 °C subjected to bacteria expressing empty vector (CTRL) or dsRNA targeting autophagy genes (**a**) *lgg-1/ATG8* and (**b**) *cup-5* from day 1 of adulthood. WT-CTRL, $N = 121$; WT-*lgg-1/ATG8* RNAi, $N = 121$; WT-*cup-5* RNAi, $N = 124$; SQST-1 OE-CTRL, $N = 108$, SQST-1 OE-*lgg-1/ATG8* RNAi, $N = 116$; SQST-1 OE-*cup-5* RNAi, $N = 129$ animals. *P*-values by log-rank test. See Supplementary Data 3 for experimental repeats. **c** Colocalization of SQST-1 and LGG-1/Atg8 in the head region of animals expressing *sqst-1p::sqst-1::gfp* and *lgg-1p::tdtomato::lgg-1*. First panel is overlay of DIC (differential interference contrast) image and GFP (SQST-1) and tdTOMATO (LGG-1/Atg8) fluorescence. Second panel is overlay of GFP (SQST-1) and tdTOMATO (LGG-1/Atg8) fluorescence, followed by single fluorescent channels. Scale bar: 10 and 2.5 μm in close-up. Intensity correlation quotient (ICQ) values between 0 and 0.5 indicate colocalization and values between 0 and −0.5 indicate segregated fluorescence. ICQ score were quantified in entire head region from two independent experiments ($N = 17$ animals). Error bars indicate 95% CI. **d** GFP::LGG-1/Atg8-positive punctae in nerve-ring neurons of WT and SQST-1 OE (*sqst-1p::sqst-1*) animals on day 1 of adulthood expressing neuronal GFP::LGG-1 (*rgef-1p::gfp::lgg-1*). Scale bar: 10 μm. **e** Histogram of GFP::LGG-1/Atg8 structures binned by their diameter in nerve-ring neurons of WT ($N = 519$ GFP::LGG-1/Atg8 structures) and SQST-1 OE ($N = 1196$ GFP::LGG-1/Atg8 structures) animals with indicated median diameter. $P < 0.0001$ by Wilcoxon signed rank test. See Supplementary Table 5. **f** GFP::LGG-1/Atg8-positive punctae in nerve-ring neurons of WT and SQST-1 OE (*sqst-1p::sqst-1*) animals on day 1 of adulthood expressing GFP::LGG-1 under its endogenous promoter (*lgg-1p::gfp::lgg-1*). Scale bar: 10 μm. **g** Autophagy-flux measurements were performed at 20 °C in WT and SQST-1 OE animals on day 1 of adulthood expressing *rgef-1p::gfp::lgg-1*. WT and SQST-1 OE animals were injected with vehicle (DMSO) or bafilomycin A1 (BafA) to block autophagy at the lysosomal acidification step. GFP::LGG-1/Atg8-positive punctae were quantified from three independent experiments (WT-DMSO, $N = 30$; WT-BafA, $N = 32$; SQST-1 OE-DMSO, $N = 31$, SQST-1 OE-BafA, $N = 32$ animals). Error bars indicate 95% CI. \*\**P* < 0.01, \*\*\*\**P* < 0.0001, by two-way ANOVA with Tukey's multiple comparisons test. Source data for (**a**), (**b**), (**c**), (**e**), and (**g**) are provided in the Source Data file.

*let-60(ts)*, and *unc-54(ts)* protein-folding mutants, since these strains allowed for co-expression of GFP-tagged SQST-1. Overexpression of SQST-1::GFP improved the movement defects of *dyn-1(ts)* and *let-60(ts)*, but not muscle-specific *unc-54(ts)* mutants at the restrictive temperature in an autophagy-dependent manner (Supplementary Fig. 11a–c). Furthermore, SQST-1::GFP accumulated upon *lgg-1/ATG8* RNAi in nerve-ring neurons and distal intestines of *dyn-1(ts)* and *let-60(ts)* mutants, but not in body-wall muscle of *unc-54(ts)* mutants, which under control conditions showed a striking increase in SQST-1::GFP accumulation (Supplementary Fig. 11d–f). These data indicate that the effects of SQST-1 on proteostasis are tissue-specific and that SQST-1 overexpression may improve proteostasis in neurons and intestine at least in part by engaging autophagy. Moreover, we found that overexpression of SQST-1::GFP in *atg-18(gk378)* hypomorphs, which display a high percentage of larval lethality[51], significantly increased the survival of *atg-18(gk378)* larvae, while overexpressing LGG-1/Atg8 did not (Supplementary Fig. 12a, b). This indicates that SQST-1-overexpressing animals could at least partially overcome the autophagy defects caused by the hypomorphic mutation in *atg-18/WIPI*. Taken together, our data support the notion that SQST-1 overexpression in *C. elegans* improves neuronal and intestinal proteostasis upon proteotoxic conditions in an autophagy-dependent manner, possibly by facilitating the elimination of protein aggregates.

**Overexpression of SQST-1 induces neuronal autophagy**. Since autophagy-related genes were required for the SQST-1-mediated benefits on lifespan and proteostasis (Fig. 3), we explored whether SQST-1 overexpression was sufficient to regulate autophagy. First, we asked whether SQST-1 co-localized with LGG-1/Atg8 in adult *C. elegans* co-expressing *sqst-1p::sqst-1::gfp* and *lgg-1p::tdtomato::lgg-1* from endogenous promoters. Most of the SQST-1::GFP structures observed in and around the nerve ring of day 1 adult animals co-localized with tdTOMATO::LGG-1 (Fig. 4c), consistent with SQST-1 interacting with LGG-1/Atg8 in adult *C. elegans*, as previously shown in embryos[52].

We next monitored autophagy in individual tissues of *C. elegans* expressing GFP-tagged LGG-1/Atg8 and overexpressing untagged SQST-1. Endogenous SQST-1 overexpression led to a marked increase in the number as well as size of GFP::LGG-1/Atg8-positive structures around the nerve ring compared with WT (Fig. 4d, e, Supplementary Fig. 12c), whereas a lipidation-deficient GFP::LGG-1(G116A) reporter showed few 'background' punctae, irrespective of SQST-1 levels (Supplementary Fig. 12c). Notably, the median diameter of neuronal GFP::LGG-1/Atg8-positive punctae was ~400 nm larger in SQST-1-overexpressing animals compared with WT animals (Fig. 4e and Supplementary Table 5), suggesting that SQST-1 overexpression potentiates an enlargement of the autophagosome in neurons.

Under basal conditions, the endogenously expressed GFP::LGG-1/Atg8 reporter does not allow for a clear separation of neurons from other tissues in the head region[53]. In SQST-1-overexpressing animals, however, enlarged GFP::LGG-1/Atg8-positive punctae were observed in the region of the nerve-ring neurons, which provides further evidence for the increased formation of enlarged autophagic structures in that tissue (Fig. 4f). We also observed an increase in the number and size of GFP::LGG-1/Atg8-positive puncta in the intestine and the pharynx of these animals (Supplementary Fig. 12d–g and Supplementary Table 5).

To further assess the dynamic status of the autophagy process, we injected SQST-1-overexpressing animals with BafA in autophagy-flux assays. The number of GFP::LGG-1/Atg8-positive punctae in neurons and the intestine of both WT and SQST-1-overexpressing animals increased upon BafA injection (Fig. 4h and Supplementary Fig. 12h), consistent with an induction of autophagy in these tissues. In contrast, the pharynx of SQST-1-overexpressing animals showed no change in the number of GFP::LGG-1/Atg8-positive punctae after BafA injection (Supplementary Fig. 12i), indicating a block in autophagy in this tissue. Thus, our results indicate that SQST-1 overexpression leads to autophagy induction in neurons and the intestine, which supports the idea that SQST-1-mediated longevity and improved proteostasis upon proteotoxic conditions occurs at least in part by the induction of tissue-specific autophagy.

## Discussion

While autophagy as a process has been linked to aging and age-related diseases, the role of selective autophagy remains less clear. Here we demonstrate an important role for the selective autophagy receptor SQST-1 in aging and proteostasis, since we found *sqst-1* to be required for the organismal benefits of a hormetic HS in *C. elegans*, and overexpression of SQST-1 was sufficient to extend lifespan and improve proteostasis in an autophagy-dependent manner. Collectively, our studies indicate that SQST-1, and possibly selective autophagy, plays an important role in longevity and health upon proteotoxic conditions.

We have previously shown that core autophagy genes, including *lgg-1/ATG8* and *bec-1/ATG6*, are required for the lifespan extension induced by hormetic HS[2] as well as by multiple conserved longevity paradigms[3]. In contrast, we found that *sqst-1* was not required for the longevity of long-lived *daf-2* insulin/IGF-1 receptor mutants, germline-less *glp-1* mutants, and animals with reduced mTOR levels (*let-363* RNAi) or mitochondrial respiration (*cyc-1* RNAi), indicating a context-specific requirement for *sqst-1* in lifespan. Since *sqst-1* is required for lifespan extension caused by reduced levels of the mitochondrial protein frataxin *frh-1*[30], which leads to iron depletion and the induction of stress responses[23,54], we propose that *sqst-1* may be specifically required in mediating selective autophagy under conditions of proteotoxic stress.

While *sqst-1* was required systemically for autophagy induction in response to a hormetic HS, loss of *sqst-1* did not abrogate PolyQ aggregation under basal conditions, which is in contrast to the loss of core autophagy genes that has been shown to increase PolyQ aggregates[2,45]. This may indicate that PolyQ aggregates in *C. elegans* are either turned over by bulk autophagy rather than *sqst-1*-mediated aggrephagy, or that bulk autophagy plays a role in preventing the formation of PolyQ aggregates possibly by the degradation of soluble PolyQ. Under basal conditions, *sqst-1* was instead specifically required for autophagosome formation in neurons. This implies that autophagy in non-neuronal tissues is mediated in an *sqst-1*-independent manner, and suggests that *sqst-1*-mediated neuronal autophagy may not be important for lifespan determination under non-stress conditions in *C. elegans*. Since *sqst-1* mutants had similar levels of PolyQ aggregation in neurons compared with WT animals, it is possible that there are *sqst-1*-dependent and independent autophagy pathways in *C. elegans* neurons. Such a possible *sqst-1*-independent autophagy pathway must, however, be *lgg-1/Atg8*-independent, since *sqst-1* mutants were essentially void of GFP::LGG-1-positive punctae. To this point, it would be of interest to investigate neuronal roles for LGG-2, another Atg8 homolog in *C. elegans*, in future experiments.

While *sqst-1* was only required upon proteotoxic conditions, we show here that endogenous overexpression as well as neuronal overexpression of an autophagy receptor, SQST-1, led to longevity at 20 °C. Furthermore, SQST-1 overexpression was sufficient to increase the mobility of neuronally expressed PolyQ proteins. This indicates that increased SQST-1 levels can change the nature of PolyQ proteins toward increased mobility, and thus possibly improved degradation. It will be interesting to determine how SQST-1 affects the biophysical properties of PolyQ proteins. One possibility could be that increased SQST-1 association with PolyQ aggregates modulates the nature of the aggregates, making them cytoprotective rather than detrimental for the cell as previously suggested[9]. Alternatively, it is possible that SQST-1 overexpression increases the degradation of PolyQ proteins, which could thus decrease the overall concentration and consequently increase the mobility of PolyQ proteins. Future experiments are needed to address these possibilities.

We found that SQST-1-overexpressing animals have reduced levels of ubiquitinated proteins, which could be due to increased degradation of ubiquitinated proteins, or due to changes in the abundance or activity of ubiquitin ligases or deubiquitinating enzymes. While we have not addressed the latter possibility, we found that SQST-1 overexpressing animals have increased proteasome activity, albeit modestly compared with germline-less *glp-1* mutants[48]. This, in combination with the increased autophagy flux in SQST-1-overexpressing animals, supports the idea that increasing SQST-1 levels increases protein degradation capabilities in *C. elegans*. Such increased protein degradation capabilities could contribute to the proteostasis and lifespan-

extending benefits observed in these animals. Importantly, we demonstrate here that SQST-1 overexpression can improve proteostasis in a *C. elegans* neurodegenerative-disease model. p62 overexpression has been shown to alleviate symptoms in an Alzheimer's disease mouse model by the induction of autophagy[10]. We show that the improved proteostasis of animals with temperature-sensitive missense aggregation mutations similarly required autophagy genes, suggesting that autophagy, at least in part, is an important effector mechanism of the beneficial effects of SQST-1 overexpression.

Consistently, we showed that endogenous SQST-1 overexpression can lead to the induction of autophagy in neurons and the intestine. This leads to the question of how SQST-1 could mediate autophagosome induction. Recent findings point to the intriguing possibility that the presence of cargo can induce the formation of the autophagic isolation membranes[55]. This is supported by findings that p62 is present at the autophagosome-formation site before the interaction with LC3/Atg8 in mouse-embryonic fibroblasts[6,56]. In addition, ubiquitinated cargo may directly contribute to the recruitment of the autophagy machinery[57], and induce p62 phase separation and cargo segregation in vitro[58]. Thus, it seems plausible that ubiquitinated proteins that accumulate upon a hormetic HS in *C. elegans*, induce autophagy via recruitment of SQST-1. This suggests the possibility, that *C. elegans* neurons are prone to accumulate increased levels of ubiquitinated proteins under basal conditions that could be cleared more efficiently upon SQST-1 overexpression. Alternatively, increased SQST-1 levels, particularly in neurons, could lead to the formation of ubiquitinated aggregates, which are somehow "primed" for removal by autophagy, causing organismal benefits.

Since SQST-1 overexpression led to the formation of unusually large autophagosomes in neurons, it is similarly interesting to speculate on how SQST-1 could mediate autophagosome size. Studies in yeast have shown that the amount of Atg8 can regulate the size of autophagosomes by controlling the expansion of the phagophore[59]. Recent in vitro reconstitution models suggest that p62 oligomerization into filaments enables the formation of large clusters with ubiquitinated proteins that at the same time can interact with the isolation membrane[55,60]. This interaction facilitates the bending of the membrane around the cargo[55,60]. Therefore, it is conceivable that increased SQST-1 protein levels could drive the formation of larger filaments and thus dictate the size of autophagosomes. It is possible that such larger autophagosomes are able to degrade ubiquitinated proteins, since SQST-1 overexpression reduced the steady-state levels of ubiquitinated proteins in *C. elegans*. Recent evidence suggests that the degradation of specific cargos by selective autophagy could be important for maintaining health[3]. It would therefore be interesting to further explore what the content of these larger autophagic structures is and their specific role in individual tissues, including in the pharynx in which SQST-1 overexpression caused a block in autophagy.

In conclusion, this study provides evidence for a central role of *sqst-1* in HS-mediated homeostasis, illustrating a specific requirement for *sqst-1* during proteotoxic stress. Moreover, overexpression of SQST-1 increased longevity and improved neuronal and intestinal proteostasis by inducing autophagy in *C. elegans*. [We note that while this paper was under revision, David Walker's group reported that adult overexpression of Ref(2)/p62 in *Drosophila* similarly extends lifespan in an autophagy-dependent manner[61].] While our data support previous observations showing that increasing p62 levels in neurodegenerative disease is beneficial in mice[10], accumulation of p62 in mammals can be cytotoxic in some tissues, and can have cancerous effects under specific circumstances[15,62]. Indeed, mammalian p62 is a

multifunctional protein involved in multiple cellular processes besides autophagy, such as several signal-transduction cascades, amino-acid sensing and the antioxidant response, that are mediated by additional protein domains not present in *C. elegans* SQST-1[63]. Notably, the cancerous effects of p62 appear to be independent of the autophagy-related functions of p62[62,64]. Nevertheless, an increase in p62-mediated selective autophagy under cytotoxic conditions could be beneficial and possibly counteract age-related neurodegenerative diseases.

## Methods

**Strains**. Strains were maintained and cultured under standard conditions at 20 °C with the exception of autophagy measurements after hormetic heat shock, when animals were cultured at 15 °C[65]. For experiments with temperature-sensitive *glp-1* (*e2144*) mutants, both *glp-1(e2144)* and wild-type (WT) N2 animals were allowed to hatch at 20 °C for 24 h and then moved to 25 °C until day 1 of adulthood. *Escherichia coli* OP50 was used as a food source unless RNAi was initiated. For RNAi experiments, animals were grown on HT115 bacteria from the time of RNAi initiation (see below). See Supplementary Data 4 for strains used in and created for this study.

**RNA interference**. Gene inactivation was achieved by feeding *C. elegans* with RNAi bacterial clones expressing double-stranded RNA (dsRNA) targeting the gene of interest. Clones were obtained from the Ahringer RNAi library (*cup-5, epg-5, lgg-1/Atg8, vps-34, cyc-1, nuo-6*) or the Vidal RNAi library (*atg-18, bec-1/ATG6*). The *let-363* and *daf-2* RNAi clones and the empty vector (L4440) for controls were provided by Drs. J. Avruch, A. Dillin, and A. Fire, respectively. All RNAi clones were verified by sequencing. For RNAi experiments, HT115 bacteria were grown in Luria-Bertani (LB) liquid culture medium containing 0.1 mg ml$^{-1}$ carbenicillin (Carb; BioPioneer) and 80 µl aliquots of bacterial suspension were spotted onto 6 cm nematode growth medium (NGM) containing 0.1 mg ml$^{-1}$ Carb (NGM/Carb plates). Bacteria were allowed to grow for 1–2 days. For induction of dsRNA expression, 80 µl of a solution containing 0.1 M isopropyl-b-D-thiogalactoside (Promega/Gold Biotechnology) and 500 mg ml$^{-1}$ Carb was placed directly onto the lawn. For two generations of RNAi, animals were synchronized by hypochlorous acid treatment or eggs were manually transferred onto NGM plates seeded with dsRNA-expressing HT115 bacteria, once these animals reached adulthood, their progeny was transferred onto NGM plates seeded with dsRNA-expressing HT115 bacteria. For whole-life RNAi, animals were synchronized by hypochlorous acid treatment or eggs were manually transferred onto NGM plates seeded with dsRNA-expressing HT115 bacteria. For adult-only RNAi, animals were synchronized by hypochlorous acid treatment and eggs were allowed to hatch on NGM plates seeded with OP50 bacteria. On day 1 of adulthood, animals were transferred to NGM/Carb plates seeded with dsRNA-expressing or control bacteria.

**Autophagy measurements**. Autophagy was monitored by counting GFP-positive LGG-1/Atg8 punctae in body-wall muscle, proximal intestinal cells, and terminal pharyngeal bulb of strain DA2123, MAH693, MAH925, MAH236, MAH864, RD202, and MAH711. Autophagy was monitored in the nerve-ring neurons of strain MAH242, MAH690, MAH944, MAH855, MAH856, MAH744, MAH779, MAH785, and MAH904. See Supplementary Data 4 for genotypes and strain information. For hormetic HS experiments, animals were raised at 15 °C and subjected to HS for 1 h at 36 °C. For all other experiments, animals were raised at 20 °C.

For autophagic flux assays, bafilomycin A1 (BafA) (BioViotica) or vehicle (dimethylsulfoxide, DMSO) was injected into *gfp::lgg-1/ATG8*-expressing animals on day 1 of adulthood[40]. BafA was resuspended in DMSO to a stock concentration of 25 µM. The co-injection dye Texas Red Dextran 3000 MW (Molecular Probes) was resuspended in water to a stock concentration of 25 mg ml$^{-1}$. A 100 µM BafA working solution was prepared by adding 1 µl of the 25 mM BafA stock solution to 249 µl of water. The 50 µM BafA injection solution was prepared by mixing 5 µl of 100 µM BafA with 1 µl of the 25 mg ml$^{-1}$ co-injection dye with 4 µl of water. A 0.2% DMSO injection solution containing the co-injection dye was prepared the same way. The BafA or DMSO injection solution was injected into the anterior intestinal area or close to the terminal pharyngeal bulb and animals were allowed to recover on NGM plates with OP50 for 2 h. Surviving animals that scored positive for the red dye were used for GFP::LGG-1 punctae quantification.

For imaging and punctae quantification, animals were mounted on a 2% agarose pad in M9 medium containing 0.1% NaN$_3$ and GFP::LGG-1/Atg8 punctae were counted using a Zeiss Imager Z1 including apotome.2 with a Hamamatsu orca flash 4LT camera and Zen 2.3 software. The total number of GFP::LGG-1/Atg8-positive punctae was counted in body-wall muscle, the three to four most proximal intestinal cells or the terminal pharyngeal bulb at ×1000 magnification. For imaging and punctae quantification of the nerve-ring neurons, Z-stack images were acquired at 1.0 µm slice intervals. At least 8–10 animals were imaged for each condition and results were combined from three independent experiments and analyzed using Student's *t*-test, one-way analysis of variance (ANOVA) or two-way ANOVA as applicable (GraphPad Prism). The size of autophagic structures was determined by tracing the outline of GFP::LGG-1/Atg8-positive punctae from acquired images using ImageJ. Histograms of diameters of autophagic structures binned by diameter were made using GraphPad Prism.

**Thermorecovery assays**. For each strain, four 6 cm NGM plates with 20–40 animals per plate were incubated in a single layer in a HERAtherm incubator (ThermoFisher) at 36 °C for 7 h. Animal survival was measured after 7 h at 36 °C followed by a 'recovery' period of 20 h at 20 °C. Survival was assessed by scoring the animals' voluntary movement. For hormetic HS, animals were incubated at 36 °C for 1 h on day 1 of adulthood and incubated for 7 h at 36 °C on day 3 of adulthood. The average percentage survival and s.e.m. were calculated and the data were analyzed by one-way analysis of variance (ANOVA) or two-way ANOVA as applicable (GraphPad Prism).

**Lifespan analysis**. Lifespan was measured at 20 °C using six 6 cm NGM or NGM/Carb plates seeded with OP50 bacteria or dsRNA-expressing bacteria with 15–20 animals per plate. The L4 larval stage was recorded as day 0 of the lifespan and animals were transferred at least every other day to new 6 cm NGM plates throughout the reproductive period. For RNAi experiments, feeding with dsRNA-expressing or control bacteria was initiated on day 1 of adulthood unless indicated otherwise. For hormetic HS, animals were incubated at 36 °C for 1 h on day 1 of adulthood. Animals were scored as dead if they failed to respond to gentle prodding with a platinum-wire pick. Censoring occurred if animals desiccated on the edge of the plate, escaped, ruptured or suffered from internal hatching. Statistical analysis was performed using Stata software (StataCorp) or Oasis 2[66]. *P*-values were calculated with the log-rank (Mantel–Cox) method. See Supplementary Tables 2–5 and Supplementary Data 1–3 for a summary of all lifespan experiments. For methodology on healthspan parameters see Supplementary Methods.

**Fluorescence recovery after photobleaching (FRAP)**. The mobility of PolyQ aggregates was assayed using FRAP on a Nikon A1R HD confocal microscope[67]. Animals were either subjected to a hormetic HS (36 °C, 1 h) on day 1 of adulthood, or maintained at control conditions (20 °C). For FRAP of neuronal Q40::YFP, animals were aged until day 8 of adulthood, for FRAP of intestinal Q44::YFP and muscle Q35::YFP, animals on day 3 of adulthood were used. For FRAP Animals were anaesthetized with M9 medium containing 0.1% NaN$_3$ and aggregated and diffuse PolyQ::YFP was bleached with 100% power of a 405 nm laser for 3.6 s (neurons) or 5.83 s (muscle and intestine). Recovery was monitored for a duration of 2 min, with measurements every 30 s (neurons) or every 10 s (muscle and intestine) with 1% of a 488 nm laser. Fluorescence intensity was relative to pre-bleach intensity and a non-bleached area as well as a non-fluorescent background was monitored and corrected for. At least 30 animals were imaged for each condition and results were combined from three independent experiments and analyzed using two-way ANOVA (GraphPad Prism).

**Analysis of PolyQ strains and temperature-sensitive mutants**. The number of neuronal PolyQ aggregates was counted in individual animals of strains AM101, MAH755, MAH876 and MAH877 (*f*+*r* on days 2–14 of adulthood. The number of intestinal PolyQ aggregates was counted in individual animals of strains MAH602 and MAH691 on days 2–5 of adulthood. The number of muscle PolyQ aggregates was counted in individual animals of strains AM140 and MAH745 on days 2–3 of adulthood. See Supplementary Data 4 for genotypes and strain information. Animals were raised and maintained at 20 °C. For hormetic HS animals were raised at 20 °C and then maintained under control conditions or subjected to HS for 1 h at 36 °C on day 1 of adulthood. Animals were imaged on NGM plates without food after anaesthetization with M9 medium containing 0.1% NaN$_3$. Images were acquired with a Leica DFC310 FX camera. Aggregates were counted using the 'Cell counter' function of ImageJ software. Averages and s.e.m. were calculated and the data were analyzed by two-way ANOVA (GraphPad Prism). For methodology on paralysis measurements of temperature-sensitive mutants, see Supplementary Methods.

**Imaging of fluorescent reporter strains**. For images of higher magnification (including for colocalization of SQST-1::GFP and tdTOMATO::LGG-1/Atg8), animals were mounted on a 2% agarose pad in M9 medium containing 0.1% NaN$_3$ and imaged on a Zeiss Imager Z1 including apotome.2 with a Hamamatsu orca flash 4LT camera and Zen 2.3 software at ×1000 magnification. For colocalization, images were analyzed using the colocalization plugin Coloc2 in Fiji-ImageJ (National Institutes of Health). The correlation between the red and green fluorescence is given as intensity correlation quotient (ICQ) values[68]. For random or mixed patterns of fluorescence intensity of the red and green channel this number will tend toward 0, for segregated localization of the red and green fluorescence signal the ICQ score will tend toward −0.5, and for colocalization of the fluorescent signals, the ICQ score will tend toward +0.5.

**Biochemical assays**. For methodology on differential detergent extraction, western blot analysis, and proteasomal activity assays, see Supplementary Methods.

**Quantitative RT-PCR.** For methodology on quantitative RT-PCR, see Supplementary Methods. For sequences of primers used for quantitative RT-PCR, see Supplementary Table 6.

**Reporting summary.** Further information on research design is available in the Nature Research Reporting Summary linked to this article.

## Data availability

The authors declare that all data supporting the findings of this study are available within this article, its Supplementary Information files, the peer-review file, or are available from the corresponding author upon reasonable request.

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

## Acknowledgements

We thank Dr. Andrew Hellman, Andrew Davis, Linnea Adams, Reza Badiee, Kylie Hooker, Leslie Boyd, and the Sanford Burnham Prebys Imaging Core for technical assistance; Stanislav Kumsta for help with data analysis; and Drs. Renaud Legouis, Hong Zhang, Kang Shen, Alicia Melendez, Cynthia Kenyon, Roger Tsien, Andrew Fire, Erik Jorgensen, and Andrew Dillin for kindly providing reagents. Some of the nematode strains used in this work were provided by the Caenorhabditis Genetics Center (University of Minnesota), which is supported by the NIH–Office of Research Infra-structure Program (P40 OD010440). This work was funded by the Deutsche For-schungsgemeinschaft and the European Research Council (consolidator grant 616499) (to T.H.), the Julie Martin Mid-Career Award in Aging Research from The Ellison Medical Foundation/AFAR (to M.H.), and by NIH grants AG058038 (to C.K.), and AG028664 (to M.H.).

## Author contributions

C.K., J.T.C., D.V., and M.H. designed the experiments. J.T.C. and E.P.T. constructed strains. C.K., J.T.C., Re.L., E.P.T., Y.Y., E.C., S.L., and M.H. performed lifespan experiments. J.T.C., Re.L., E.C., S.L., and M.H. contributed to healthspan experiments. C.K. and Re.L. performed stress assays, aggregation assays and western blots. Ru.L. and I.S.M. performed proteostasis experiments. C.K. performed all other experiments. C.K. and M. H. wrote the paper, with input from coauthors. The SQST-1 antibody was provided by A. S. and T.H. ahead of publication[33].

## Competing interests

The authors declare no competing interests.
