## [Peer Review File · Nature Communications]

Reviewers' comments:

Reviewer #1 (Remarks to the Author):

Kumsta et al present in this study an analysis of the overexpression of SQST-1/p62 and its effect on hormesis, lifespan and proteostasis. This is a follow-up study of their previous publication (Kumsta et al., 2017) that showed that SQST-1/p62 mRNA levels increase upon a beneficial heat shock. The authors attempted now to shed light onto the underlying mechanism. The authors show a low of data, yet they are not coherently presented and are not always convincing. A lot of controls are missing and they tend to overinterpret their data. Their insight into selective autophagy is limited to one factor, SQST-1/p62. Yet p62 is functionally involved in autophagy as well as the proteolytic targeting and turnover by the proteasome. Their findings rely mainly on GFP reporters and descriptive assays. There is no conceptual advance to gain mechanistic insights. The quality of the imaging data and in particular their quantification is not sufficient.

Major comments:

1. Quantifications of SQST-1/p62 are only based on imaging of GFP-fusion proteins and should be complemented by western blot.
2. How much overexpression of SQST-1/p62 relative to endogenous levels did the authors achieve? It will certainly make a difference if the levels differ by 2 or 10-fold. The assessment via Western blot using a SQST-1-specific antibody is therefore a must.
3. The statement on p8 l110 "... suggesting that sqst-1/p62 and by extension selective autophagy ..." requires an analysis of more genes of the selective autophagy pathway.
4. A control is missing that demonstrates that the fusion protein SQST-1-GFP is functional like the untagged SQST-1.
5. Figure 1 a+b: ubiquitin is not only a degradation signal for autophagy. The involvement of the UPS is not even mentioned, let alone analyzed. Controls for recovery in the presence of inhibitors for UPS (epoxomicin or MG132) as well as autophagy (3-MA etc) are missing.
6. Figure 1f: a brightfield image is required to see which area of the nematode is imaged. E.g. show the entire head. The signal is very fuzzy and does not resemble the usual Q40-YFP aggregation pattern of those nematodes. In addition, to actually assess the effect on polyQ aggregation: FRAP or filter retardation assays are necessary to detect and more importantly to quantify a decrease or increase in aggregation.
7. Figure 5: It is absolutely essential to show the data for both, permissive as well as non-permissive temperature for any assay that uses ts mutants.
8. Figure S7f: neuronal overexpression (again: how much? – and transcript levels rarely correlate with protein levels) of SQST-1 leads to a pronounced reduction in the number of offspring. Why is that?
9. Figure 4f+g and p16 l 313: "...endogenous and neuronal overexpression of SQST-1/p62 was sufficient to reduce the levels of ubiquitinated proteins. ... suggests that SQST-1/p62 overexpression can facilitate the degradation of ubiquitinated proteins." An alternative explanation is a reduction in E3 ligase abundance or activity that would also lead to reduced Ub-levels. Free Ubiquitin is also reduced. How do the authors explain that? And why did they show separate blots?
10. Figure S1c: why is the GFP antibody signal for SQST-1 Δ 113-377::GFP so weak compared to SQST-1::GFP? These transgenic lines are not comparable if the difference in expression is like it seems 100-fold.
11. Figure S1e: why is the sqst-1 mutant more mobile than the wt?

Minor comments:

1. Please change LGG/1/LC3 to LGG-1/LC3

2. Figure S1e-g: why are there only error bars for selected data points?

Reviewer #2 (Remarks to the Author):

Hormetic heat shock induces autophagy and improves proteostasis in an autophagy dependent manner in *C. elegans*. In this elegant study, Kumsta and colleagues demonstrated that the *C. elegans* p62 homolog, SQST-1, is required for the benefits of a hermetic heat shock and that SQST-1 is required for autophagy induction after heat stress. The authors further showed that SQST-1 overexpression leads to tissue-specific induction of autophagy, resulting in an extension of lifespan and an improvement of proteostasis. The experiments are well-designed, the data are with high quality, and the manuscript is well-written. It is suitable for publication in *Nature Communication* after the following minor concerns are addressed.

1. Several places describing the autophagy pathway in the introduction part need be clarified.
 - a. Page 3, line 4, "Upon completion, autophagosomes fuse with acidic lysosomes.." Autophagosomes undergo a maturation process by fusing with vesicles originating from endolysosomal compartments before forming degradative autolysosomes.
 - b. Page 3, line 16, "..in which cytosolic proteins are indiscriminately recycled.." bulk autophagy degrades proteins as well as other cytoplasmic contents.
 - c. Page 4, lines 34-36, "..These aggregates are significantly reduced in mice and flies when p62 is deleted, indicating a role for p62 in inclusion-body formation." "These aggregates" should be specified. Inclusion-body may not be an appropriate term, as protein aggregates in all tissues are referred here.
2. The authors used the name "sqst-1/p62" throughout the manuscript. It is better to use "sqst-1".
3. LGG-1 is the GABARAP homolog. The LC3 homolog in *C. elegans* is encoded by *Igg-2*. The authors could simply indicate that LGG-1 is a homolog of ATG8 family proteins. Both "LGG-1/LC3" and "LGG-1/LC3" are used in the manuscript. "LGG-1" is more appropriate.
4. Page 9, line 154 and many other places, LC3/Atg8 labels autophagic structures at all stages, including isolation membranes, autophagosomes, amphisomes and autolysosomes. It is more appropriate to state LGG-1 puncta as autophagic structures.
5. The authors showed that sqst-1 is required for basal autophagy in neurons. However, the number of PolyQ aggregates is unchanged in nerve-ring neurons in sqst-1 mutants. The presence of both sqst-1-dependent and sqst-1-independent basal autophagy pathways in neurons should be clearly stated. Is sqst-1 required for autophagy in neurons at the embryonic stage? Is sqst-1 required for induced autophagy in the nerve-ring such as in mTOR signaling inactivation mutant animals?
6. Page 13, line 252, "cup-5/MCOLN1, which is involved in lysosomal fusion". cup-5 is not required for lysosomal fusion. It is required for regeneration of lysosomes from endo/auto/lysosomes.
7. Page 17, lines 346-348, "..and no further increase in SQST-1/p62::GFP fluorescence upon *Igg-1/LC3* RNAi". This gives an impression that SQST-1 is not degraded by autophagy in muscle cells. The authors should indicate that high expression levels of SQST-1::GFP exceed the degradation capability of autophagy in muscle cells.

Reviewer #3 (Remarks to the Author):

In a previous study, Kumsta et al., reported that hormetic heat stress activates autophagy to extend lifespan and promote proteostasis in the nematode *C. elegans*. Here they expanded their investigation to further explore the links between autophagy, hermetic heat shock and proteostasis, and found that sqst-1/p62, a known component of the autophagy mechanism, plays key roles in the regulation of lifespan and proteostasis.

They report that sqst-1/p62 is needed for the benefit of hermetic heat stress, as it is required for

the induction of autophagy, mainly in nerve ring neurons. They also found that the over expression of sqst-1/p62, in all tissues or exclusively in neurons, is sufficient to extend lifespan in an autophagy-dependent manner and to improve proteostasis of worms that express metastable proteins.

Based on these results they propose that hermetic heat shock extends lifespan and promotes proteostasis, at least partially, by inducing the expression of sqst-1/p62.

This study is potentially interesting and timely, however several important issues have to be addressed in order to strengthen and solidify the conclusions.

1. Whilst the authors claim that the levels of p62/SQST-1 affect the aggregation of polyQ-YFP proteins (for instance in Fig. 1, E-F), they only count fluorescent foci. This is not an appropriate technique to compare the rate of aggregation, as the number of foci is not necessarily correlated with the level of aggregation. To properly compare rates of aggregation they have to homogenize the worms, separate soluble from insoluble fraction by high speed centrifugation (using an ultracentrifuge) and compare the levels of polyQ-YFP in the different fractions by Western blot analysis. Alternatively, they can use the technique that was developed by the Nollen lab, to visualize polyQ-YFP assemblies in agarose gels (Holmberg and Nollen *Methods Mol Biol.* 2013;1017:193-9.).

2. The apparent discrepancy between the observation that sqst-1/p62 is needed for the benefits that are conferred by hermetic heat shock (Fig. 1), to the nearly identical induction of autophagy related genes and hsp-70 in wild-type and ok2892 worms (supplemental Fig. 5, A-B) is puzzling, and should be better explained.

3. In supplemental Fig. 3 the authors compare the number of LGG-1 puncta in worms that express GFP::LGG-1 under the regulation of the rgef-1 promoter to these that were observed in worms that express GFP::LGG-1 (G116A) mutant, which is driven by the lgg-1 promoter. How can they distinguish between differences that may stem from the mutation to these which could emanate from differences in expression levels? Please explain.

4. In Fig. 4, F-G they show that the over-expression of sqst-1/p62 endogenously or in neurons reduce the levels of protein ubiquitination. Since the degradation of ubiquitinated proteins is also promoted by proteasomes, it is crucial to test whether worms that over-express sqst-1/p62 exhibit elevated proteasome activity. This can be measured in-vitro by commercial substrates.

5. In supplemental table 3 the authors present results which show that daf-2 RNAi extends the lifespan of ok2892 worms and conclude that sqst-1/p62 is dispensable for the longevity mechanism downstream of the insulin/IGF signaling cascade. To further assess the possible roles of sqst-1/p62 as a regulator of lifespan it would be important to test whether the knockdown of this gene abrogates the longevity phenotypes that stem from dietary restriction (for instance in strain AD1116). In addition, they should test if sqst-1/p62 is needed for the longevity of worms that carry mutations which reduce the activity of the electron transport chain.

6. The discussion comprehensively describes the potential roles of sqst-1/p62 in hermetic heat stress and proteostasis. One important aspect of the relations between the heat shock response and proteostasis is missing. Several research groups have shown that the ability to activate the heat shock response comes at the expense of proteostasis. First, the Morimoto lab reported that inactivating the AFD thermosensory neurons by knocking down gcy-8, renders the heat shock response inactive but at the same time provides protection from the aggregation of polyQ-YFP proteins (Prahlad and Morimoto, *PNAS* 2011). In addition, the Cohen lab found that knocking down the expression of the neuronal gene nhl-1 sensitizes the worms to heat stress but protects them from proteotoxicity (Volovik et al., *Cell Reports* 2014). Adding a paragraph that discusses how sqst-1/p62 may be involved in the complex relations between stress resistance and proteostasis will enhance the manuscript.

Response to reviewers for manuscript NCOMMS-18-35744

We enclose our revised manuscript entitled “The autophagy receptor p62/SQST-1 promotes proteostasis and longevity in *C. elegans* by inducing autophagy” by Kumsta *et al.* We appreciate the reviewers’ valuable comments, which we have addressed point-by-point below. In response to this feedback, we have made the following major changes and experimental additions:

1. To better illustrate the novelty and impact of our paper, we have restructured the results section to better emphasize our key finding that SQST-1 overexpression induces autophagy and improves proteostasis in a tissue-specific manner in *C. elegans*. To this end, the main figures now focus on neurons as a key tissue.
2. We have quantified the levels of SQST-1 overexpression in our integrated SQST-1-overexpressing strains, *i.e.*, GFP-tagged SQST-1 under control of both the endogenous and a neuronal promoter and found that the overexpression generally was increased by >20 fold. See Reviewer 1, point 1 and 7 for full discussion and new **Supplementary Figure 6**.
3. To strengthen our conclusion that SQST-1 protein is responsible for the extension of lifespan we observe in *C. elegans*, we constructed and overexpressed a truncated SQST-1 mutant, lacking the ubiquitin-binding domain (SQST-1 Δ UBA). Notably, we found that SQST-1 Δ UBA was not sufficient to extend lifespan in *C. elegans*. See Reviewer 1, point 3 for full discussion and new **Figure 3b and Supplementary Table 4**.
4. We have performed FRAP experiments to characterize the mobility of PolyQ aggregates. Importantly we found that a hormetic heat shock increased the mobility of neuronal Q40 aggregates in an *sqst-1*-dependent manner. See Reviewer 1, point 5 for full discussion and **Figure 2e-f and Supplementary Fig. 5**.
5. We have added new experiments demonstrating that SQST-1 overexpression increased the lifespan of animals expressing neuronal PolyQ proteins and increased the mobility of neuronal PolyQ proteins, similarly to a hormetic heat shock. These findings indicate that SQST-1 levels can affect the properties of neuronal PolyQ proteins with beneficial effects for the animal (**Figure 3g-h and Supplementary Table 7**).
6. We have performed proteasomal activity assays of SQST-1-overexpressing animals and found that SQST-1-overexpressing animals have modestly elevated proteasome activity, in addition to induced autophagic flux. See Reviewer 1, point 4 for full discussion and **Supplementary Figure 8**.
7. We have analyzed additional conserved longevity models for their requirement for *sqst-1*. Our results show that *sqst-1* was not required for the lifespan of these longevity paradigms and may indicate that selective *sqst-1*-mediated autophagy is required under specific, potentially proteotoxic conditions, such as a hormetic heat shock. See Reviewer 3, point 5 for full discussion and **Supplementary Table 3**.
8. We have now included diameter measurements of autophagic structures in SQST-1-overexpressing animals (**Figure 4e and Supplementary Figure 10e-f**).

With these new insights, as well as several other advancements highlighted below, the revised manuscript has been significantly strengthened.

Novel sections in the revised manuscript are indicated in blue text.

Reviewer 1:

Kumsta et al present in this study an analysis of the overexpression of SQST-1/p62 and its effect on hormesis, lifespan and proteostasis. This is a follow-up study of their previous publication (Kumsta et al., 2017) that showed that SQST-1/p62 mRNA levels increase upon a beneficial heat shock. The authors attempted now to shed light onto the underlying mechanism.

The authors show a low of data, yet they are not coherently presented and are not always convincing. A lot of controls are missing and they tend to overinterpret their data. Their insight into selective autophagy is limited to one factor, SQST-1/p62. Yet p62 is functionally involved in autophagy as well as the proteolytic targeting and turnover by the proteasome. Their findings rely mainly on GFP reporters and descriptive assays. There is no conceptual advance to gain mechanistic insights. The quality of the imaging data and in particular their quantification is not sufficient.

We have taken the reviewer's comments to heart and have added several controls and novel data to strengthen our conclusions (see summary above). In particular, our novel finding that overexpression of a selective autophagy receptor can increase lifespan and induce autophagy with benefits on proteostasis contributes to our understanding of *sqst-1*-mediated selective autophagy upon heat stress and its role in aging and proteostasis. Our findings that *sqst-1* is specifically required upon proteotoxic heat stress is the first step in revealing the identity of specific and potentially cytotoxic cargos relevant to aging and age-related diseases.

Major comments:

1. Quantifications of SQST-1/p62 are only based on imaging of GFP-fusion proteins and should be complemented by western blot. How much overexpression of SQST-1/p62 relative to endogenous levels did the authors achieve? It will certainly make a difference if the levels differ by 2 or 10-fold. The assessment via Western blot using a SQST-1-specific antibody is therefore a must.

We have now performed Western Blot analysis to quantify the amount of SQST-1 overexpression in the transgenic strains in our study. To compare the levels of SQST-1::GFP to endogenous SQST-1, we used CRISPR-Cas9 to endogenously tag *sqst-1* with GFP (*sqst-1(syb940[sqst-1::gfp])*). The levels of SQST-1::GFP in this strain were, however, so low that no signal was detected on the Western blot using an antibody against GFP. When we subjected these animals to several generations of *lgg-1* RNAi (to ensure that autophagy was equally and fully blocked in all strains analyzed, causing SQST-1 accumulation), we were able to detect endogenous SQST-1::GFP. Using this approach, the levels of GFP-tagged SQST-1 expressed under control of the endogenous promoter was found to be 20-fold or more. Similarly, GFP-tagged SQST-1 under control of the neuronal was 20-fold or more (**Supplementary Fig. 6b**).

We also used an SQST-1 antibody provided by Dr. Thorsten Hoppe (which was recently published, Springhorn and Hoppe, Methods Enzymol. 2019). However, in this complementary approach, we found that the levels of SQST-1 in WT animals were not consistently detected; we therefore again used the *sqst-1(syb940[sqst-1::gfp])* strain for comparison and found that GFP-tagged SQST-1 expressed under control of the endogenous promoter was about 30-fold. Interestingly, GFP-tagged SQST-1 under control of the neuronal was 60-fold or more (**Supplementary Fig. 6c**). We have added a note in the figure legend of **Supplementary Fig. 6** that SQST-1 expression levels in WT and SQST-1 overexpressing animals are prone to fluctuation, presumably because of the constant degradation of SQST-1 by autophagy.

Additionally, we used the SQST-1-specific antibody to quantify the levels of untagged SQST-1 and a newly generated untagged SQST-1 mutant (SQST-1 Δ UBA) expressed from extrachromosomal arrays in *sqst-1(syb764)* deletion mutants. The *sqst-1(syb764); Ex[sqst-1p::sqst-1]* animals had 3-fold overexpression of SQST-1, which was sufficient to extend their lifespan, while *sqst-1(syb764); Ex[sqst-1p::sqst-1 Δ UBA]* had an accumulation of SQST-1 of about 6-fold, but were not long-lived (**Supplementary Fig. 6d**). Overall, our efforts to measure SQST-1 overexpression showed that overexpression of SQST-1 within a rather broad range of overexpression can extend *C. elegans* lifespan.

2. The statement on p8 l110 "... suggesting that *sqst-1*/p62 and by extension selective autophagy ..." requires an analysis of more genes of the selective autophagy pathway.

While the study of a selective autophagy receptor like *sqst-1* implicitly studies selective autophagy, we have edited the manuscript and now only state that *sqst-1* is required, rather than selective autophagy.

Nevertheless, we tested whether the limited number of selective autophagy receptors that have been identified thus far in *C. elegans* were, like *sqst-1*, transcriptionally increased upon a hormetic heat shock. The selective autophagy receptor *sepa-1* is important for the removal of P-granules during development (Zhang Y. et al., Cell, 2009). We found that the expression levels of *sepa-1* during adulthood were hardly detectable (**Figure R1A**) and its transcript levels were not increased upon a hormetic heat shock (**Figure R1B**). The transcript levels of mitophagy receptor *dct-1*/BNIP3 (Palikaras K. et al., Nature, 2015) were reduced upon a hormetic heat shock (**Figure R2B**), as has been previously shown (Brunquell J. et al. BMC Genomics 2016). Lastly, the autophagy receptor *allo-1* is important for the clearance of paternally provided mitochondria (allophagy) in the *C. elegans* embryo (Sato M. et al, Nat Cell Biol. 2018). A role for *allo-1* in adult stages has not been described, and we did not find *allo-1* transcript levels to be upregulated upon a hormetic heat shock (**Figure R2B**). Therefore, *sqst-1* remains the only, as of yet, identified selective autophagy receptor that is upregulated upon a hormetic heat shock.

Figure R1: mRNA expression analysis of selective autophagy receptors after a hormetic heat shock.

(A) GFP expression of animals expressing *sepa-1p::sepa-1::gfp* on day 1 of adulthood. Exposure time, 1000 ms. Scale bar, 200 μ m. * indicates embryos in-utero.

(B) Transcript levels of genes of selective autophagy receptors on day 1 of adulthood maintained under control conditions (CTRL) or subjected to heat shock for 1 h at 36°C (HS). Data are the mean \pm s.e.m. of three biological replicates, each with three technical replicates, and were normalized to the mean expression levels of levels of housekeeping genes *nhr-23*, *ama-1*, and *cdc-42*. ns: $P > 0.05$, ** $P < 0.01$, **** $P < 0.0001$, by two-way ANOVA with Tukey's multiple comparisons test.

3. A control is missing that demonstrates that the fusion protein SQST-1-GFP is functional like the untagged SQST-1.

We note that the original version of our manuscript included lifespan data demonstrating that overexpression of untagged SQST-1, similar to GFP-tagged SQST-1, increased longevity (**Supplementary Table 4**). Irrespectively, we have conducted additional experiments to strengthen our conclusions that overexpression of SQST-1 (untagged and tagged) extends lifespan: We created constructs for the expression of untagged SQST-1 Δ UBA, in which the ubiquitin-binding domain (UBA) was deleted, and injected these constructs into our newly created CRISPR-knockout *sqst-1*(*syb764*) mutants, to prevent any interaction with endogenous *sqst-1*. Lifespan analysis of these strains (**Fig. 3b** and **Supplementary Table 4**) demonstrated that overexpression of SQST-1 Δ UBA did not extend lifespan, while overexpression of untagged SQST-1 in *sqst-1*(*syb764*) mutants extended lifespan to a similar extent as GFP-tagged SQST-1

overexpression in a wild-type background. We confirmed the expression of SQST-1 and SQST-1 Δ UBA in *sqst-1(syb764)* mutants using Dr. Hoppe's SQST-1-specific antibody (**Supplementary Fig. 6**). Taken together, these results emphasize that overexpression of SQST-1, either as a tagged or non-tagged protein, extend lifespan, and that the lifespan extension require the UBA domain. The increased interaction of SQST-1 with ubiquitinated proteins upon SQST-1 overexpression is therefore likely required for the lifespan-extending effects of SQST-1 overexpression.

4. Figure 1 a+b: ubiquitin is not only a degradation signal for autophagy. The involvement of the UPS is not even mentioned, let alone analyzed. Controls for recovery in the presence of inhibitors for UPS (epoxomicin or MG132) as well as autophagy (3-MA etc) are missing.

As the reviewer rightly points out, ubiquitination can serve as a signal for proteasomal degradation as well as autophagy, which we mentioned both in the introduction as well as in the discussion of the original manuscript.

In Figure 1a-b of the previous version of this manuscript, we illustrated that heat shock leads to the increased high molecular weight ubiquitin conjugates to corroborate the findings from human cells. This figure was used as an introductory figure and additional premise for our analysis of the role of SQST-1 in the benefits of a hormetic heat shock. Since we do not draw any conclusions from this figure on the involvement of autophagy, the UPS or deubiquitination, we feel that this figure stands well on its own. In the restructuring of the manuscript, we have, however, moved this figure into the supplementary information as **Supplementary Fig. 1** to better reflect the supportive nature of our current data set.

To the request for experimental investigations, we have analyzed SQST-1 overexpression strains for proteasomal activity using an *in vitro* assay in collaboration with Dr. David Vilchez at CECAD in Cologne, Germany. We found that animals overexpressing SQST-1 under control of the endogenous promoter had marginally increased proteasome activity (10%, SQST-1 OE(1), $P > 0.05$ by One-way ANOVA) and significantly increased proteasome activity (30%, SQST-1 OE(2) $P < 0.01$ by One-way ANOVA), while animals neuronally overexpressing SQST-1 had modestly increased activity (15%, Neuronal SQST-1 OE(1), $P > 0.05$ by One-way ANOVA). Since all SQST-1-overexpressing strains showed at least a trend towards increased proteasome activity, it was not elevated to the level of *glp-1* (100%, $P < 0.0001$ by One-way ANOVA), which we used as a positive control for increased proteasomal activity. We therefore conclude that SQST-1-overexpressing animals have modestly elevated proteasome activity (**Supplementary Fig. 8**). This modestly increased proteasomal activity could contribute to the improved proteostasis of SQST-1-overexpressing animals, we have therefore expanded our discussion of a potential involvement of the proteasome in the introduction, results and discussion.

5. Figure 1f: a brightfield image is required to see which area of the nematode is imaged. E.g. show the entire head. The signal is very fuzzy and does not resemble the usual Q40-YFP aggregation pattern of those nematodes. In addition, to actually assess the effect on polyQ aggregation: FRAP or filter retardation assays are necessary to detect and more importantly to quantify a decrease or increase in aggregation.

As requested by the reviewer we have now replaced the images in former Figure 1f with images taken on a Zeiss Z1 imager and have also included DIC images of the head region. These images are in new **Fig. 2c**.

Furthermore, we implemented and performed FRAP and Filter-trap experiments to characterize the morphology, mobility and solubility of YFP-tagged PolyQ proteins in different *C. elegans* tissues, *i.e.*, nerve-ring neurons, intestine and body-wall muscle. We have added the FRAP experiments to the manuscript but were unable to characterize PolyQ aggregation using Filter-trap analyses. Our detailed findings are outlined below:

FRAP experiments

1. Neuronal PolyQ: We have performed FRAP experiments (see Methods section for experimental details) and found that a hormetic heat shock increased the mobility of neuronal PolyQ aggregates in an *sqst-1*-dependent manner, without changing the mobility of the diffusely expressed protein (**Fig. 2. e-f and**

Supplementary Fig. 5c). To determine whether SQST-1 overexpression mimics hormetic heat shock conditions, we expressed untagged SQST-1 in animals expressing neuronal PolyQ. FRAP analysis of PolyQ proteins in animals overexpressing SQST-1 showed that SQST-1 overexpression increased the mobility of both the aggregated as well as the diffuse portion of neuronal PolyQ proteins, similarly to a hormetic heat shock. This is a significant finding as it indicates that SQST-1 levels can affect the properties of neuronal PolyQ proteins in a beneficial way. Moreover, we found that SQST-1 overexpression could substantially increase the lifespan of animals expressing neuronal PolyQ proteins without changing the number of aggregates. We have included these novel experiments in **Fig. 3g-h and Supplementary Table 7.**

2. Intestinal PolyQ: We did not find any changes in the mobility of PolyQ proteins expressed in the intestine after a hormetic heat shock (**Supplementary Fig. 5f**). We do, however, note that intestinal aggregates were much more mobile than aggregates in other tissues (~50% fluorescence recovery to pre-bleach fluorescence levels, compared with ~17% for muscle and ~35% of neurons). It is therefore possible that a hormetic heat shock changes the level of intestinal PolyQ proteins (aggregated or diffuse) by a different mechanism than in neurons. It will be interesting to determine whether SQST-1 overexpression can change the mobility of intestinal PolyQ aggregates similar to neuronal PolyQ (We could not easily generate animals overexpressing intestinal PolyQ and overexpressing SQST-1, since they both carry the same co-injection marker). Since we were unable to further characterize intestinal PolyQ proteins in *sqst-1* mutants with and without heat shock at this point, we have changed the focus of our manuscript on SQST-1 function to neuronal tissues and have moved our analyses on intestinal PolyQ to the supplements (**Supplementary Fig. 5**).
3. Muscle PolyQ: We did not find any changes in the mobility of PolyQ proteins expressed in body-wall muscle after a hormetic heat shock (**Supplementary Fig. 5h**). This was expected, since we did not observe any benefits of hormetic heat shock on PolyQ aggregation in the muscle; in fact, we detected more PolyQ aggregates in body-wall muscles after a hormetic heat shock. This could indicate that muscles may be unable to mount the same beneficial responses to a hormetic heat shock as the other tissues, which is in line with our observation that SQST-1 overexpression did not improve the protein-folding defects in muscle-specific temperature-sensitive folding mutants (*unc-54(ts)*).

Filter-trap experiments

1. Neuronal PolyQ: We have performed filter trap experiments according to Sin O et al. Mol Cell 2017 and Bioprotocols 2018, in which they analyzed muscle-specific PolyQ aggregates by filter trap. For our analyses, we used wild-type and *sqst-1(ok2892)* mutants expressing neuronal Q40 proteins, which is to the best of our knowledge the first attempt of trapping neuronal PolyQ proteins of this length in *C. elegans*. The animals were subjected to a hormetic heat shock or maintained under control conditions. We harvested about ~200 μ l of animals on day 8 of adulthood to ensure the formation of a large number of aggregates, since neuronal aggregates are small. We were, however, unable to detect aggregates by Filter trap, even though we were able to visually see aggregates (>10 per animal in the nerve-ring neurons) and capture Q40::YFP in these samples using SDS-Page. A possible reason that we were unable to trap neuronal PolyQ proteins is presumably due to the small number of neuronal aggregates compared to total protein (**Figure R2A**, see next page). Further experiments optimizing this protocol, by first isolating neuronal cells for instance, are needed and we were unable to establish such new techniques in time for this resubmission.
2. Intestinal PolyQ: We used the same protocol mentioned above (Sin O et al. Mol Cell 2017 and Bioprotocols 2018) that was established for muscle-specific PolyQ aggregates, to trap intestinal PolyQ proteins, which again is, to the best of our knowledge, the first attempt of trapping intestinal PolyQ proteins in *C. elegans*. We harvested animals at day 5 of adulthood after visually inspecting them for aggregate formation (>45 aggregates per animal). We were unable to trap intestinal PolyQ aggregates, because they were surprisingly soluble in 2% SDS. We additionally tried to trap intestinal PolyQ aggregates in 0.5% SDS but were not able to consistently trap the aggregates, even though we captured Q44::YFP in these samples using SDS-Page. This finding is consistent with our FRAP experiments that showed that PolyQ aggregates were more mobile (~50% fluorescence recovery to

pre-bleach fluorescence levels, compared with ~17% for muscle and ~35% for neurons) and thus presumably more soluble than aggregates in muscle or neurons (**Figure R2B-C**). Further experiments are necessary to optimize this protocol, possibly by further aging the animals to ensure more profound aggregation, we were unable to establish such new techniques in time for this resubmission.

3. **Muscle PolyQ:** We used muscle PolyQ aggregates as a control in our initial experiments, as they have been previously shown to be easily trapped using this assay (Sin O et al. Mol Cell 2017). We have, however, not performed any further Filter-trap assays with animals expressing muscle PolyQ proteins, since neither a hormetic heat shock nor *sqst-1* mutation had any effects on muscle PolyQ aggregates (**Figure R2B**).

In sum, our FRAP experiments have led to novel results, namely that a hormetic heat shock increased mobility of neuronal PolyQ aggregates in an *sqst-1*-dependent manner, and that SQST-1 overexpression similarly increased the mobility of neuronal PolyQ proteins. We have included all of our FRAP experiments in the new manuscript in **Fig. 1d-e** and **Supplementary Fig. 5**. We were unable to analyze tissue-specific PolyQ proteins using Filter trap at this point. Further optimization is necessary as outlined above.

Figure R2: Filter-trap assays of animals expressing tissue-specific PolyQ aggregates and subjected to heat shock.

Filter trap of crude-protein lysates from PolyQ-expressing animals grown on media plates containing 5-Fluoro-2'-deoxyuridine (FUdR) to prevent progeny production. The presence of aggregates was confirmed by microscopy before lysis. At least 200 μ l of animals were collected per biological sample and disrupted using a motorized pestle for seven cycles of 20 s with 5 min rest in between cycles. Lysates were resuspended in FTA sample buffer (10 mM Tris-Cl pH 8.0, 150 mM NaCl, 2% SDS) with protease inhibitors (Complete, Roche) To detect SDS-insoluble aggregates, 100 μ g of total protein was mixed with 1 M DTT and FTA sample buffer and 25 μ g of total protein plus two five-fold serial dilutions were filtered through a 0.22 micron cellulose acetate membrane using a 48-well Bio-Dot microfiltration system (Bio-Rad). Proteins were blocked for 30 min with 5% milk in TBS-T. PolyQ antibodies were detected using antibodies against GFP (α -GFP).

(A) Filter-trap assay with 3-fold serial dilutions of crude protein extract from 2 biological samples of wild-type (WT) and *sqst-1(ok2892)* animals expressing neuronal PolyQ collected on day 8 of adulthood and were either heat shocked for 1h at 36°C (HS)

on day 1 of adulthood or kept at control conditions (CTRL) (20°C). PolyQ::YFP was detected in 3 µg of lysates using SDS-PAGE and immunoblotting against GFP and Tubulin (after stripping). Two more biological samples showed similar results.

(B) Filter-trap assay with 3-fold serial dilutions of crude protein extract from wild-type animals expressing intestinal PolyQ or muscle PolyQ collected on day 4 of adulthood. Soluble Q40::YFP was detected in 3 µg of lysates using SDS-PAGE and immunoblotting against GFP.

(C) Filter-trap assay with 3-fold serial dilutions of crude protein extract from three biological samples wild-type (WT) and *sqst-1(ok2892)* animals expressing intestinal PolyQ collected on day 5 of adulthood and were either heat shocked for 1h at 36°C (HS) on day 1 of adulthood or kept at control conditions (CTRL) (20°C). Lysates were resuspended in FTA sample buffer containing only 0.5% SDS. PolyQ::YFP was detected in 3 µg of lysates using SDS-PAGE and immunoblotting against GFP and Actin (after stripping). This experiment was first attempted using three biological samples using 2% SDS.

6. Figure 5: It is absolutely essential to show the data for both, permissive as well as non-permissive temperature for any assay that uses ts mutants.

We have analyzed the paralysis phenotypes of temperature-sensitive mutant animals expressing SQST-1::GFP at the permissive temperature. As expected, the temperature-sensitive folding mutants did not show any movement defects at 15°C and did not accumulate SQST-1::GFP at 15°C. These experiments are shown in **Supplementary Fig. 9**.

7. Figure S7f: neuronal overexpression (again: how much? – and transcript levels rarely correlate with protein levels) of SQST-1 leads to a pronounced reduction in the number of offspring. Why is that?

It is indeed interesting that the neuronal but not the endogenous SQST-1-overexpressing animals have a reduced number of progeny. This could indicate a possible trade-off between longevity and progeny production in animals overexpressing SQST-1 in neurons. Furthermore, it raises an interesting question whether neuronal SQST-1-overexpressing animals extend lifespan by a different mechanism than animals overexpressing SQST-1 endogenously. To this point, we have included the following statement in the figure legend of **Supplementary Fig. 7** (please also see response to Reviewer 3 point 6 below):

“We note that SQST-1 overexpression under the control of the pan-neuronal promoter but not the endogenous promoter lead to a decrease in the number of progeny production, indicating a possible trade-off between longevity and progeny production in animals overexpressing SQST-1 in neurons.”

As mentioned above (under point 1), we have quantified the expression levels of the neuronal SQST-1-overexpressing strain using Western Blot analysis to be >20-fold.

8. Figure 4f+g and p16 | 313: “...endogenous and neuronal overexpression of SQST-1/p62 was sufficient to reduce the levels of ubiquitinated proteins. ... suggests that SQST-1/p62 overexpression can facilitate the degradation of ubiquitinated proteins.” An alternative explanation is a reduction in E3 ligase abundance or activity that would also lead to reduced Ub-levels.

The reviewer’s point on alternative explanations on the reduction of ubiquitinated proteins in SQST-1 overexpressing animals is well taken. Our newly added proteasome activity assays (see point 4) suggest that SQST-1-overexpressing animals have modestly elevated proteasome activity (**Supplementary Fig. 8**). In light of our additional data on the proteasome capacity of SQST-1-overexpressing strains, we have now expanded our discussion of a potential involvement of the proteasome in the discussion, where we say on p.19:

“We found that SQST-1-overexpressing animals have reduced levels of ubiquitinated proteins, which could be due to increased degradation of ubiquitinated proteins, or due to changes in the abundance or activity of ubiquitin ligases or deubiquitinases. While we have not addressed the latter possibility, we found that SQST-1 overexpressing animals have increased proteasome activity, albeit the increase is modest compared to germline-less *glp-1* mutants⁵⁵. This, in combination with the increased autophagy flux in SQST-1-overexpressing animals, supports the idea that increasing SQST-1 levels increases protein degradation capabilities in *C. elegans*. Such increased protein degradation capabilities could contribute to the proteostasis and lifespan-extending benefits observed in these animals.”

Free Ubiquitin is also reduced. How do the authors explain that? And why did they show separate blots?

We thank the reviewer for pointing out this error in the manuscript. We are in fact not detecting free ubiquitin, but mono-ubiquitinated proteins. We have now corrected this error in the manuscript. We are not showing separate blots, but rather different lanes from the same blot as the lanes we wish to present were not loaded next to each other (we have made a clarifying comment in the figure legend). We have included the image of the entire blot here for the referee's review (**Figure R3**).

Figure R3: SQST-1-overexpressing animals have reduced ubiquitinated proteins.

Western blot of wild-type (WT), X (undisclosed strains), SQST-1-overexpressing (*sqst-1p::sqst-1::gfp*) (SQST-1 OE) and neuronal SQST-1-overexpressing (*rgef-1p::sqst-1::gfp*) (Neuronal SQST-1 OE) animals on day 1 of adulthood using antibodies against ubiquitin, indicating high molecular weight (HMW) ubiquitin, and tubulin.

9. Figure S1c: why is the GFP antibody signal for SQST-1 Δ 113-377::GFP so weak compared to SQST-1::GFP? These transgenic lines are not comparable if the difference in expression is like it seems 100-fold.

We have now removed previous Supplementary Figure 1a-c, and instead cite a recently published paper that established that *sqst-1(ok2892)* mutants express a truncated SQST-1 protein (Springhorn and Hoppe, Methods Enzymol. 2019). See also comment to point 1.

10. Figure S1e: why is the *sqst-1* mutant more mobile than the wt?

We have now added healthspan measurements of a full *sqst-1(syb764)* deletion to exclude that the expression of truncated SQST-1 Δ 113-377 in *sqst-1(ok2892)* mutants is responsible for the increased thrashing of *sqst-1(ok2892)* mutants. Notably, *sqst-1(syb764)* mutants also exhibited increased thrashing compared to wild-type animals. While we currently have no explanation for this phenotype, we could speculate that the reduced number of GFP::LGG-1/Atg8-positive punctae in neurons of *sqst-1* mutants, implying reduced levels of *sqst-1*-dependent autophagy, could change the innervation or neurotransmitter release required for locomotion. More experiments are needed to address this point and would be interesting to carry out in the future.

Minor comments:

1. Please change LGG-1/LC3 to LGG-1/LC3

We thank the reviewer for pointing out this error, which we have corrected in the revised manuscript.

2. Figure S1e-g: why are there only error bars for selected data points?

In the previous version of this manuscript, we displayed error bars as SEM and since the SEM was very small, only error bars greater than the data point symbol were displayed. We have now changed the error bars to the 95% CI and for most data points the error bars are now displayed on the graph. We describe the error bars in the figure legends and explain any possible absence of error bars therein.

Reviewer 2:

Homeostatic heat shock induces autophagy and improves proteostasis in an autophagy dependent manner in *C. elegans*. In this elegant study, Kumsta and colleagues demonstrated that the *C. elegans* p62 homolog, SQST-1, is required for the benefits of a homeostatic heat shock and that SQST-1 is required for autophagy induction after heat stress. The authors further showed that SQST-1 overexpression leads to tissue-specific induction of autophagy, resulting in an extension of lifespan and an improvement of proteostasis. The experiments are well-designed, the data are with high quality, and the manuscript is well-written. It is suitable for publication in Nature Communication after the following minor concerns are addressed.

We thank the reviewer for the positive comments on our manuscript.

1. Several places describing the autophagy pathway in the introduction part need be clarified.
 - a. Page 3, line 4, "Upon completion, autophagosomes fuse with acidic lysosomes.." Autophagosomes undergo a maturation process by fusing with vesicles originating from endolysosomal compartments before forming degradative autolysosomes.

We thank the reviewer for this valuable feedback, and we have now edited this section as follows:

"Upon completion, the so-called autophagosome matures and can form amphisomes with vesicles from the endolysosomal compartments. Autophagosomes or amphisomes then fuse with acidic lysosomes,..."

- b. Page 3, line 16, "...in which cytosolic proteins are indiscriminately recycled.." bulk autophagy degrades proteins as well as other cytoplasmic contents.

We have now changed 'cytosolic proteins' to 'cytosolic components'.

- c. Page 4, lines 34-36, "...These aggregates are significantly reduced in mice and flies when p62 is deleted, indicating a role for p62 in inclusion-body formation." "These aggregates" should be specified. Inclusion-body may not be an appropriate term, as protein aggregates in all tissues are referred here.

In general, protein aggregates can be defined by their non-native secondary structure, poor solubility in solutions, and aberrant subcellular or extracellular localization. The term inclusion body has been used to describe intracellular foci or granules in which aggregated proteins are sequestered into. This occurs either through direct deposition and self-assembly of non-native protein monomers into growing polymers or by the clustering of misfolded protein aggregates into larger aggregate structures. Since the nature of the p62-positive aggregates is as of yet unclear (especially in *C. elegans*), we will refrain from the term 'inclusion body' and just use the term 'aggregate' or 'ubiquitin-positive aggregate' (when appropriate) to be more precise.

2. The authors used the name "sqst-1/p62" throughout the manuscript. It is better to use "sqst-1".

Since we now only use *C. elegans* and yeast nomenclature we have changed *sqst-1/p62* to "*sqst-1*" throughout the manuscript.

3. LGG-1 is the GABARAP homolog. The LC3 homolog in *C. elegans* is encoded by *lgg-2*. The authors could simply indicate that LGG-1 is a homolog of ATG8 family proteins. Both “LGG-1/LC3” and “LGG/1/LC3” are used in the manuscript. “LGG-1” is more appropriate.

We thank the reviewer for pointing out the typographical error (LGG/1/LC3). Since we now only use *C. elegans* and yeast nomenclature, we have changed LGG-1/LC3 to LGG-1/Atg8 throughout the manuscript. We prefer to add the additional name to help non-*C. elegans* readers.

4. Page 9, line 154 and many other places, LC3/Atg8 labels autophagic structures at all stages, including isolation membranes, autophagosomes, amphisomes and autolysosomes. It is more appropriate to state LGG-1 puncta as autophagic structures.

We have now made the following changes in the manuscript to address this comment:

- When we first introduce the GFP-tagged LGG-1/Atg8 reporter (now p.6, line 97) we say “a GFP-tagged LGG-1/Atg8 reporter, which visualizes autophagic structures (i.e., pre-autophagosomal structures, autophagosomes and amphisomes) as fluorescent punctae.”
- When we describe our results, we refer to these structures as GFP::LGG-1/Atg8-positive punctae
- When we discuss our results, we refer to these punctae as “autophagic structures”.

5. The authors showed that *sqst-1* is required for basal autophagy in neurons. However, the number of PolyQ aggregates is unchanged in nerve-ring neurons in *sqst-1* mutants. The presence of both *sqst-1*-dependent and *sqst-1*-independent basal autophagy pathways in neurons should be clearly stated.

It is possible that, as the reviewer suggests, there are *sqst-1*-dependent and independent autophagy pathways in *C. elegans* neurons. Since there are virtually no GFP::LGG-1/Atg8-positive punctae in *sqst-1* mutants (**Fig. 1e**, formerly Fig. 2e), an *sqst-1*-independent autophagy pathway must also be *lgg-1*-independent. Since there are only two Atg8 homologs in *C. elegans*, an *sqst-1/lgg-1*-independent autophagy pathway would most likely require *lgg-2*. Since GFP::LGG-2 is expressed in the neurons of adult *C. elegans* (Alberti et al., Autophagy, 2010), it would be interesting to address the interaction of SQST-1 and LGG-2 in neurons in the future, as neuronal-specific reagents for LGG-2 are as of yet unavailable. We have now added this section in the discussion on p. 18:

“Since *sqst-1* mutants had equivalent numbers of neuronal PolyQ aggregates, it is possible that there are *sqst-1*-dependent and independent autophagy pathways in *C. elegans* neurons. Such a possible *sqst-1*-independent autophagy pathway must however be *lgg-1/Atg8*-independent, since *sqst-1* mutants are void of GFP::LGG-1-positive punctae. To this point, it would be of interest to investigate neuronal roles for LGG-2, another Atg8 homolog in *C. elegans*, in future experiments.”

Alternatively, it is possible that neuronal autophagy does not play a role in the degradation of neuronal PolyQ aggregates. We have previously shown that inhibition of autophagy genes using RNAi causes increased accumulation of PolyQ aggregates in neurons of *C. elegans* (Jia K et al. Autophagy 2007, Kumsta et al. Nature Comm, 2017). In this previous study of ours, autophagy genes were reduced in the entire body of *C. elegans*, and not specifically in the neurons. *C. elegans* neurons are largely refractory to RNAi, therefore the inhibition of autophagy in non-neuronal tissues appears to contribute to the accumulation of neuronal PolyQ aggregates in *C. elegans*. We show that *sqst-1* mutants were not impaired for autophagy in non-neuronal tissues (**Fig. 1**, formerly Fig. 2), and this may be the reason why the numbers of neuronal PolyQ aggregates are not increased in *sqst-1* mutant animals. It is an intriguing speculation that signals from non-neuronal tissues could affect neuronal PolyQ aggregation, which should be further tested.

Is *sqst-1* required for autophagy in neurons at the embryonic stage?

We agree with the reviewer that this is an interesting question. We have used the neuronal *rgef-1* promoter to assess neuronal autophagy (*rgef-1p::gfp::lgg-1*) and *rgef-1* expression can be detected in *C. elegans* embryos from the comma stage on, but not in earlier embryos. To address the reviewer’s question, we

imaged wild-type and *sqst-1* embryos expressing *rgef-1p::gfp::lgg-1* and detected few GFP::LGG-1-positive punctae in embryos in both genetic backgrounds, indicating that *sqst-1* may not be required for GFP::LGG-1 punctae formation during embryogenesis (**Figure R4**). We are aware that a direct quantification of punctae in these embryos would help support this idea; however, this has proven to be challenging, because the embryos are motile at these stages and resistant to any anesthetic we have tried (data not shown). Further optimization for these experiments would therefore be necessary, but since our manuscript focuses on the adult function of SQST-1 on autophagy, we believe such experiments would be beyond the scope of this manuscript.

Figure R4: *sqst-1* is not required for GFP::LGG-1 punctae formation in neuronal tissues during embryogenesis. Embryos of wild-type and *sqst-1(syb764)* animals expressing *rgef-1p::gfp::lgg-1* were imaged at the indicated embryonic stage. GFP::LGG-1-positive punctae are indicated with white arrowheads.

Is *sqst-1* required for induced autophagy in the nerve-ring such as in mTOR signaling inactivation mutant animals?

We thank the reviewer for this question. While it has been shown that the number of GFP::LGG-1-positive punctae is increased in hypodermal seam cells of *C. elegans* larvae with reduced mTOR pathway activity (Hansen *et al.*, 2008, and Robida-Stubbs *et al.*, 2012), the autophagy status in other tissues, including neurons, upon mTOR reduction has not yet been assessed. We have now monitored GFP::LGG-1 punctae in wild-type and *sqst-1(ok2892)* adult animals upon *mTOR/let-363* inhibition. We found that *let-363* RNAi increased the number of GFP::LGG-1-positive punctae in nerve-ring neurons, consistent with an induction of neuronal autophagy. *sqst-1* mutants, however, did not exhibit an increased number of GFP::LGG-1-positive punctae in nerve-ring neurons upon *let-363* RNAi (**Figure R5**). This indicates that *sqst-1* is required for neuronal autophagy upon *let-363* inhibition, similar to the requirement we report in animals subjected to hormetic heat shock. Interestingly, *sqst-1* was not required for the lifespan extension induced

by inhibition of *let-363* (**Supplementary Table 3**) (see Reviewer 3, point 5 below), which is in contrast to a hormetic heat shock. Our results suggest that *sqst-1*-mediated neuronal autophagy may not be important for lifespan determination under basal conditions in *C. elegans*. This finding is of particular interest to us, as we are addressing the tissue-specific requirements for selective as well as general autophagy in ongoing studies.

Figure R5: *sqst-1* is required for GFP::LGG-1 punctae formation upon mTOR inhibition. Wild-type and *sqst-1*(*syb764*) animals expressing *rgef-1p::gfp::lgg-1* were raised on bacteria expressing either empty vector (CTRL) or dsRNA targeting *let-363* (mTOR) (*let-363* RNAi) until day 1 of adulthood. GFP::LGG-1-positive punctae were quantified in three independent experiments. Error bars indicate s.e.m. ns: P>0.05, ****P<0.0001, by two-way ANOVA.

6. Page 13, line 252, “cup-5/MCOLN1, which is involved in lysosomal fusion”. cup-5 is not required for lysosomal fusion. It is required for regeneration of lysosomes from endo/auto/lysosomes.

We thank the reviewer for pointing out this mistake. We have now changed this sentence to:

“cup-5/MCOLN1, whose loss impairs degradation of autolysosomal contents, ...”

7. Page 17, lines 346-348, “..and no further increase in SQST-1/p62::GFP fluorescence upon Igg-1/LC3 RNAi”. This gives an impression that SQST-1 is not degraded by autophagy in muscle cells. The authors should indicate that high expression levels of SQST-1::GFP exceed the degradation capability of autophagy in muscle cells.

We have now substantially revised and shortened this section. To make our point clearer, we now say in the result section on p.13:

“These data indicate that the effects of SQST-1 on proteostasis are tissue-specific, *i.e.*, SQST-1 overexpression may improve proteostasis in neurons and intestine at least in part by autophagy, but not in body-wall muscle, which could be either due to excessive accumulation of SQST-1::GFP that exceeds the autophagic degradation capability or blocked autophagy in muscles of *unc-54(ts)* mutants at 25°C.”

Reviewer 3:

In a previous study, Kumsta et al., reported that hormetic heat stress activates autophagy to extend lifespan and promote proteostasis in the nematode *C. elegans*. Here they expanded their investigation to further explore the links between autophagy, hermetic heat shock and proteostasis, and found that *sqst-1/p62*, a known component of the autophagy mechanism, plays key roles in the regulation of lifespan and proteostasis. They report that *sqst-1/p62* is needed for the benefit of hermetic heat stress, as it is required for the induction of autophagy, mainly in nerve ring neurons. They also found that the over expression of *sqst-1/p62*, in all tissues or exclusively in neurons, is sufficient to extend lifespan in an autophagy-dependent manner and to improve proteostasis of worms that express metastable proteins.

Based on these results they propose that hermetic heat shock extends lifespan and promotes proteostasis, at least partially, by inducing the expression of *sqst-1/p62*.

This study is potentially interesting and timely, however several important issues have to be addressed in order to strengthen and solidify the conclusions.

We thank the reviewer for the suggestions, and we believe that our new data (summarized on p.1) have strengthened our manuscript.

1. Whilst the authors claim that the levels of p62/SQST-1 affect the aggregation of polyQ-YFP proteins (for instance in Fig. 1, E-F), they only count fluorescent foci. This is not an appropriate technique to compare the rate of aggregation, as the number of foci is not necessarily correlated with the level of aggregation.. To properly compare rates of aggregation they have to homogenize the worms, separate soluble from insoluble fraction by high speed centrifugation (using an ultra-centrifuge) and compare the levels of polyQ-YFP in the different fractions by Western blot analysis. Alternatively, they can use the technique that was developed by the Nollen lab, to visualize polyQ-YFP assemblies in agarose gels (Holmberg and Nollen Methods Mol Biol. 2013;1017:193-9.).

As outlined in response to Reviewer 1, point 5 above, we have performed FRAP and Filter-trap experiments to characterize the morphology, mobility and solubility of YFP-tagged PolyQ proteins in different *C. elegans* tissues, *i.e.*, nerve-ring neurons, intestine and body-wall muscle. Please see detailed response above.

2. The apparent discrepancy between the observation that *sqst-1/p62* is needed for the benefits that are conferred by hormetic heat shock (Fig. 1), to the nearly identical induction of autophagy related genes and *hsp-70* in wild-type and *ok2892* worms (supplemental Fig. 5, A-B) is puzzling, and should be better explained.

While *sqst-1* is required for the benefits of a hormetic heat shock, including autophagosome formation, *sqst-1* is not required for the transcriptional regulation of autophagy genes or heat shock response gene *hsp-70* after a hormetic heat shock. This is an important finding because it indicates that the induction of *sqst-1* expression after heat shock is likely downstream of the transcriptional response induced by a hormetic heat shock. We have now edited this section of the results as follows on p.8:

“This suggests that *sqst-1(ok2892)* mutants can induce a heat-shock response and the concomitant increase in transcript levels of autophagy genes. Therefore, it seems likely that *sqst-1* is required downstream of the transcriptional regulation of autophagy genes induced by heat shock.”

3. In supplemental Fig. 3 the authors compare the number of LGG-1 puncta in worms that express GFP::LGG-1 under the regulation of the *rgef-1* promoter to these that were observed in worms that express GFP::LGG-1 (G116A) mutant, which is drive by the *lgg-1* promoter. How can they distinguish between differences that may stem from the mutation to these which could emanate from differences in expression levels? Please explain.

We thank the reviewer for pointing out this typographical error. In **Supplementary Fig. 2a** (formerly Supplementary Fig. 3a), we only compared GFP::LGG-1-positive punctae driven by the pan-neuronal *rgef-1* promoter. We have now corrected this mistake.

4. In Fig. 4, F-G they show that the over-expression of *sqst-1/p62* endogenously or in neurons reduce the levels of protein ubiquitination. Since the degradation of ubiquitinated proteins is also promoted by proteasomes, it is crucial to test whether worms that over-express *sqst-1/p62* exhibit elevated proteasome activity. This can be measured in-vitro by commercial substrates.

As outlined in response to Reviewer 1, point 4 above, we have now added proteasome-activity assays, which suggest that SQST-1-overexpressing animals have modestly elevated proteasome activity

(**Supplementary Fig. 8**) compared to *glp-1* animals, which exhibit elevated proteasomal activity. In light of our additional data on the proteasome capacity of SQST-1-overexpressing strains, we have expanded our discussion of a potential involvement of the proteasome. Specifically, on page 19 of the discussion, we say:

“We found that SQST-1-overexpressing animals have reduced levels of ubiquitinated proteins, which could be due to increased degradation of ubiquitinated proteins, or due to changes in the abundance or activity of ubiquitin ligases or deubiquitinases. While we have not addressed the latter possibility, we found that SQST-1 overexpressing animals have increased proteasome activity, albeit the increase is modest compared to germline-less *glp-1* mutants⁵⁵. This, in combination with the increased autophagy flux in SQST-1-overexpressing animals, supports the idea that increasing SQST-1 levels increases protein degradation capabilities in *C. elegans*. Such increased protein degradation capabilities could contribute to the proteostasis and lifespan-extending benefits observed in these animals.”

- In supplemental table 3 the authors present results which show that *daf-2* RNAi extends the lifespan of *ok2892* worms and conclude that *sqst-1/p62* is dispensable for the longevity mechanism downstream of the insulin/IGF signaling cascade. To further assess the possible roles of *sqst-1/p62* as a regulator of lifespan it would be important to test whether the knockdown of this gene abrogates the longevity phenotypes that stem from dietary restricted (for instance in strain AD1116). In addition, they should test if *sqst-1/p62* is needed for the longevity of worms that carry mutations which reduce the activity of the electron transport chain.

We agree that it is an important question to investigate whether *sqst-1* is similar to core autophagy genes in regards to being required for lifespan extension in other autophagy-dependent longevity models. Therefore, we have now analyzed additional conserved longevity mutants for their requirement for *sqst-1*, namely, germline-less *glp-1* mutants, animals with reduced mTOR levels (*let-363* RNAi) (in lieu of *eat-2* mutants, which we were unable to fully analyze in time for the resubmission), and animals with reduced mitochondrial respiration (*cyc-1* RNAi) (**Figure R6** for representative experiments). We are presenting these data in **Supplementary Table 3** of the revised manuscript. Our results show that *sqst-1* is not required for the lifespan of these longevity paradigms, indicating that selective *sqst-1*-mediated autophagy is selectively required under specific, potentially proteotoxic conditions, such as a hormetic heat shock.

Figure R6: *sqst-1* is not required for extended lifespan of several conserved longevity paradigms.

(a) Lifespan analysis of wild-type, *sqst-1(ok2892)*, *daf-2(e1370)*, and *daf-2(e1370); sqst-1(ok2892)* animals. WT compared to *daf-2*: $P < 0.0001$, *sqst-1* compared to *daf-2*; *sqst-1*: $P < 0.0001$ by log-rank test. See **Supplementary Table 3** for experimental details and repeats.

(b) Lifespan analysis of wild-type, *sqst-1(ok2892)*, *glp-1(e2144)*, and *glp-1(e2144); sqst-1(ok2892)* animals. WT compared to *glp-1*: $P < 0.0001$, *sqst-1* compared to *glp-1*; *sqst-1*: $P < 0.0001$ by log-rank test. See **Supplementary Table 3** for experimental details and repeats.

(c) Lifespan analysis of wild-type and *sqst-1(ok2892)* (*sqst-1*) animals subjected to bacteria expressing empty vector (CTRL) or dsRNA targeting *let-363* from day 1 of adulthood. WT - CTRL compared to WT – *let-363* RNAi: P<0.0001, *sqst-1* – CTRL compared to *sqst-1* – *let-363* RNAi: P<0.0001 by log-rank test. See **Supplementary Table 3** for experimental repeats.

(d) Lifespan analysis of wild-type and *sqst-1(ok2892)* (*sqst-1*) animals subjected to bacteria expressing empty vector (CTRL) or dsRNA targeting *cyc-1* from hatching on. WT - CTRL compared to WT – *cyc-1* RNAi: P<0.0001, *sqst-1* – CTRL compared to *sqst-1* – *cyc-1* RNAi: P<0.0001. See **Supplementary Table 3** for experimental repeats.

6. The discussion comprehensively describes the potential roles of *sqst-1/p62* in hermetic heat stress and proteostasis. One important aspect of the relations between the heat shock response and proteostasis is missing. Several research groups have shown that the ability to activate the heat shock response comes at the expense of proteostasis. First, the Morimoto lab reported that inactivating the AFD thermosensory neurons by knocking down *gcy-8*, renders the heat shock response inactive but at the same time provides protection from the aggregation of polyQ-YFP proteins (Pralhad and Morimoto, PNAS 2011). In addition, the Cohen lab found that knocking down the expression of the neuronal gene *nhl-1* sensitizes the worms to heat stress but protects them from proteotoxicity (Volovik et al., Cell Reports 2014). Adding a paragraph that discusses how *sqst-1/p62* may be involved in the complex relations between stress resistance and proteostasis will enhance the manuscript.

We were intrigued by the reviewer’s point, which to our understanding is the question whether the improved proteostasis in SQST-1 overexpression could come at the expense of heat stress resistance. We have therefore performed thermo-recovery experiments of SQST-1-overexpressing animals and surprisingly found, in more than six total experiments, that neuronal SQST-1-overexpressing animals, but not animals overexpressing SQST-1 from the endogenous promoter, consistently displayed increased stress resistance (**Figure R6**). In addition to this thermotolerance phenotype, neuronal SQST-1 overexpressing animals have reduced brood size, are long-lived, have improved lifespan in *dyn-1(ts)* mutant background, displayed reduced levels of ubiquitinated proteins, and had modestly increased proteasome activity. This is in line with our hypothesis that neuronal SQST-1 overexpression is sufficient to improve proteostasis. Since animals overexpressing SQST-1 under the endogenous promoter did not display increased thermo-tolerance, it is possible that neuronal SQST-1-overexpressing animals may extend lifespan by a different mechanism than animals endogenously overexpressing SQST-1. We are very interested in further investigating the differences in the phenotypes of animals endogenously and neuronally overexpressing SQST-1, but to streamline the discussion (and keep within the word limit), we prefer to keep the discussion focused on the experiments we are including in this manuscript.

Figure R6: Neuronal SQST-1-overexpressing animals have increased thermotolerance.

Survival of wild-type (WT), SQST-1-overexpressing (*sqst-1p::sqst-1::gfp*) (SQST-1 OE (1) (2)) and neuronal SQST-1-overexpressing (*rgef-1p::sqst-1::gfp*) (Neuronal SQST-1 OE (1) (2)) incubated for 7 h at 36°C on day 3 of adulthood. (N=84-105 animals, n=4 plates). Error bars indicate s.e.m. ns: P>0.05, *P<0.05 by one-way ANOVA with Tukey’s multiple comparisons test. This experiment has been repeated at least six times with similar results.

Reviewers' comments:

Reviewer #1 (Remarks to the Author):

The author's response to my main comment regarding the level of expression of the SQST-1 transgenic lines (in neurons or under the control of the endogenous promoter) is not satisfying.

They failed to detect the SQST-1 levels on a western blot even using SQST-1 antibodies provided by the Hoppe lab.

1. Why is that?

2. They engineered a sqst-1 line that carries a GFP fusion at the genetic locus. How can the authors be sure that this protein is still functional and behaves like the wild type protein?

They should analyze the autophagic flux to assess the functionality of this transgenic line.

Moreover, as they still failed to detect SQST-1::GFP using GFP antibodies they subjected the nematodes to RNAi against Igg-1 for several generation.

This is not physiological!

and clearly does not answer my initial request to assess the protein levels.

I wanted to see how much SQST-1 is produced in the wild type (N2) versus their transgenic lines.

Please do not use any tricks to somehow elevate the levels to then draw conclusions of 20-fold increases. These are very artificial set-ups.

The only untagged sqst-1 variant the authors used was not integrated. Thus again no assessment of the expression level is possible of a population of nematodes by western blot.

Regarding the polyQ aggregation - the FRAP experiments provide insight into the mobility within the aggregates. The data depicted in figure 2d and e show that the wt control shows a recovery of about 35% and wt + HS as well as sqst-1 +/-HS of about 50% - so it seems that the wt control worms cope the least with polyQ aggregates. This is not expected at all.

The Filter trap experiments failed.

Reviewer 3 actually suggested similar experiments that I fully support. A supernatant pellet assay or an SDD-AGE assay would allow to assess the polyQ aggregation.

But first - the authors need to establish in a convincing manner their nematode lines if they actually have generated an SQST-1 overexpression line or not.

Without that one cannot interpret the presented data.

Reviewer #2 (Remarks to the Author):

My concens have been satisfactorily addressed. It is acceptable now.

Reviewer #3 (Remarks to the Author):

The authors have comprehensively addressed the critique that was raised by all referees and, in

my opinion, significantly enhanced the manuscript. I therefore recommend acceptance.

Further comments or Reviewer #3 via email:

1. The referee is unsatisfied with the answers that were provided by the authors regarding his concerns of uncontrolled expression levels of p62/SQST-1. While the authors could not detect the protein using the antibody that was provided by the Hoppe lab, they created a GFP tagged construct and followed its expression levels. Furthermore, they inhibited autophagy by lgg-1 RNAi for "several generations" in order to stabilize the protein. The referee claims that these are not physiological conditions. I agree here, but think that the over-expression of any gene is not physiological yet, this technique is commonly used to investigate the biological roles of gene products. Nevertheless, the authors can better address this critique. Since p62/SQST-1 is an unstable protein, it is not sufficient to use only 800 worms for a WB analysis (as stated in the legend of supplemental figure 6). I suggest that they will upscale the number of worms, using at least 3000-4000 worms/lane (despite the fact that the gene is not integrated) and blot p62/SQST-1 again with the Hoppe lab's antibody. This experiment should be performed using worms that were either treated with lgg-1 RNAi (for four generations) or left untreated.
2. The referee raised an additional issue asking whether the p62/SQST-1::GFP protein is functional and suggests that the authors would follow the autophagic flux. This suggestion is doable and will support the conclusions of the manuscript.
3. Finally, the referee asks the authors to better analyze the rate of polyQ aggregation. The FRAP experiment improves the analysis that was presented at the first version of the manuscript. Conducting ONE of the suggested experiments to measure the rate of aggregation, NAGE or a sup-pellet sedimentation (using a polyQ antibody) will not require much effort and better establish the manuscript.

Response to reviewers for manuscript NCOMMS-18-35744A

We enclose our revised manuscript entitled “The autophagy receptor p62/SQST-1 promotes proteostasis and longevity in *C. elegans* by inducing autophagy” by Kumsta *et al.* In response to critiques from Reviewer 1 and Reviewer 3’s evaluation, we have now performed additional experiments and further strengthened our conclusions, as summarized here:

1. We have used our SQST-1-specific antibody for additional quantifications of the levels of SQST-1 overexpression in wild-type (WT) animals as well as in integrated SQST-1-overexpressing strains. The data from these experiments support our previous conclusions regarding the fold of SQST-1 overexpression in different independent lines.
2. We have now clarified the use of autophagy flux assays in SQST-1-overexpressing and the endogenously GFP-tagged strain, showing that the SQST-1 protein expressed in these transgenic lines is subject to autophagic turnover, indicating that these tagged proteins are functional.
3. We have performed Differential Detergent Extraction experiments to further characterize the effects of a hormetic heat shock on neuronal PolyQ aggregates in wild-type animals and *sqst-1* mutants and find that a hormetic heat shock increases the soluble portion of PolyQ aggregates in WT animals, but not in *sqst-1* mutants. Collectively, we have now provided four independent lines of evidence that a hormetic heat shock improves proteostasis in an *sqst-1* dependent manner in animals expressing neuronal Q40:
 1. **Number of aggregates:** Hormetic heat shock reduces the number of neuronal Q40 aggregates in an *sqst-1*-dependent manner.
 2. **Lifespan:** Hormetic heat shock increases the lifespan of animals with neuronal PolyQ aggregates in an *sqst-1*-dependent manner.
 3. **FRAP:** Hormetic heat shock increases the mobility of neuronal PolyQ aggregates in wild-type animals but not in *sqst-1* mutant animals.
 4. **Differential Detergent Extraction:** Hormetic heat shock increases the soluble portion of PolyQ aggregates and decreases the high-molecular-weight species of PolyQ aggregates in WT animals but not in *sqst-1* mutants.

With these additional experiments and changes, we have significantly strengthened the revised manuscript.

Novel sections in the revised manuscript are indicated in blue text.

Below are our detailed responses to the reviewer’s comments (highlighted in blue text).

Reviewer #1 (Remarks to the Author):

The author's response to my main comment regarding the level of expression of the SQST-1 transgenic lines (in neurons or under the control of the endogenous promoter) is not satisfying. They failed to detect the SQST-1 levels on a western blot even using SQST-1 antibodies provided by the Hoppe lab.

1. Why is that?

We apologize if this point was not clear in the previous manuscript – **Supplementary Figure S7d** (former Supplementary Figure S6c) shows an annotated band for endogenously, untagged SQST-1 in the presented Western blot along with a brief note in the figure legend that we were not able to consistently detect SQST-1 in lysates from WT animals.

We believe that the key to this problem was that the signal for SQST-1 reached saturation in lysates from SQST-1 overexpressing animals, leaving the SQST-1 signal of WT animals very faint. To address this issue and consistently measure SQST-1 levels in WT animals, we have modified our protocol for Western blot analysis by reducing the number of SQST-1-overexpressing animals that are loaded onto the gel, while maintaining the number for WT animals, and are now able to directly and consistently compare the SQST-1 levels in WT and SQST-1-overexpressing animals on the same blot. This new analysis provided reproducible results (in six new experiments) and is included as **Supplementary Figure S7c**. Similarly to previous experiments, we found that GFP-tagged SQST-1 expressed from the endogenous promoter showed a ~30-100 fold increase of SQST-1 protein levels compared to WT. Interestingly, GFP-tagged SQST-1 under control of the neuronal promoter was ~130-fold or more. Taken together, our data support the main conclusion of our manuscript, i.e., that overexpression of SQST-1 in a broad range, provides benefits to *C. elegans*.

2. They engineered a *sqst-1* line that carries a GFP fusion at the genetic locus. How can the authors be sure that this protein is still functional and behaves like the wild type protein? They should analyze the autophagic flux to assess the functionality of this transgenic line. Moreover, as they still failed to detect SQST-1::GFP using GFP antibodies they subjected the nematodes to RNAi against *lgg-1* for several generation. This is not physiological! and clearly does not answer my initial request to assess the protein levels. I wanted to see how much SQST-1 is produced in the wild type (N2) versus their transgenic lines. Please do not use any tricks to somehow elevate the levels to then draw conclusions of 20-fold increases. These are very artificial set-ups.

We indeed performed autophagy flux experiments with the CRISPR-engineered strain expressing genomically GFP-tagged SQST-1. While basal levels of SQST-1::GFP were not detectable with an anti-GFP antibody (**Figure R1** (next page), formerly Supplementary Figure S6), we observed that *lgg-1* RNAi treatment increased SQST-1::GFP levels, so that they could be detected using an anti-GFP antibody in Western blot analysis (**Figure R1a**, formerly Supplementary Figure S6b). Furthermore, *lgg-1* RNAi treatment increased the number of GFP-positive structures in SQST-1::GFP-expressing animals using fluorescence microscopy (**Figure R1b**, image was included in the first version of the manuscript). From these experiments, we conclude that GFP-tagged SQST-1 is turned over by autophagy, similar to untagged SQST-1 (Wu F. et al. Mol Cell, 2015). This indicates that GFP::SQST-1 is functional in that it can serve as a substrate for autophagy.

Figure R1: Autophagy flux assays in genomically GFP-tagged SQST-1 animals.

(a) Western blot of genomically GFP-tagged SQST-1 animals (*sqst-1(syb940[sqst-1::gfp])*) fed bacteria expressing either empty vector (CTRL) or dsRNA targeting autophagy-related gene *lgg-1/Atg8* (*lgg-1* RNAi) for four generations to block autophagy. ~800 animals were washed of multiple media plates. Antibodies against GFP (α -GFP) and actin (α -Actin) were used.

(b) GFP expression in genomically GFP-tagged SQST-1 animals (*sqst-1(syb940[sqst-1::gfp])*) fed bacteria expressing either empty vector (CTRL) or dsRNA targeting autophagy-related gene *lgg-1/Atg8* (*lgg-1* RNAi) for two generations to block autophagy. Arrows pointing to SQST-1::GFP-positive dots. Scale bar 10 μ m for close-up of head region. The CTRL image was part of former Supplementary Figure 6a, the *lgg-1* RNAi image was included in the first version of this manuscript).

We have however decided to remove this reagent and data from the manuscript since we were able to compare SQST-1 levels in our overexpressing lines to WT animals using the SQST-1-specific antibody in our new Western Blots (see above point), and we do not use this strain for other experiments in this study.

For further discussion of autophagy flux assays for SQST-1::GFP-overexpressing strains, please see response to Reviewer 3 point 2 below.

3. The only untagged *sqst-1* variant the authors used was not integrated. Thus again no assessment of the expression level is possible of a population of nematodes by western blot.

We have assessed the expression levels of untagged SQST-1 in animals carrying extrachromosomal arrays in **Supplementary Figure S7d**. Only transgenic animals were used in this analysis. While variability in expression from extrachromosomal arrays is a well-known issue, this is a commonly used approach in the field.

4. Regarding the polyQ aggregation - the FRAP experiments provide insight into the mobility within the aggregates. The data depicted in figure 2d and e show that the wt control shows a recovery of about 35% and wt + HS as well as *sqst-1* +/-HS of about 50% - so it seems that the wt control worms cope the least with polyQ aggregates. This is not expected at all.

The Filter trap experiments failed.

Reviewer 3 actually suggested similar experiments that I fully support. A supernatant pellet assay or an SDD-AGE assay would allow to assess the polyQ aggregation.

But first - the authors need to establish in a convincing manner their nematode lines if they actually have generated an SQST-1 overexpression line or not. Without that one cannot interpret the presented data.

In regards to the FRAP experiments: We agree with the reviewer that it is an important point to directly compare the mobility of neuronal PolyQ proteins in WT animals and *sqst-1* mutants under basal conditions. When we plotted WT and *sqst-1* mutants together, we observed slightly increased mobility for both aggregated and diffuse neuronal Q40, but these data sets were not statistically different (**Figure R2**). In addition to statistical notes in the figure legends, we explicitly comment on this in the manuscript text line on p10, where we say:

“Similar to aggregate load (**Fig. 2c**), the baseline values for WT and *sqst-1* animals were not significantly different in FRAP (**Fig. 2d-e**) and differential detergent extraction experiments (**Supplementary Fig. 5b**).”

Figure R2: Fluorescence recovery after photobleaching of neuronal Q40 in WT and *sqst-1* mutants.

Fluorescence-intensity measurements of the aggregated (a) and diffuse portion (b) of neuronal PolyQ (*rgef-1p::Q40::yfp*) on day 8 of adulthood in wild-type (WT) and *sqst-1(ok2892)* (*sqst-1*) animals. Fluorescence signal was bleached after 20 s and fluorescence recovery was measured for 2 min. (N=30-32, n=3). Error bars indicate 95% CI. All measurements are P>0.05 by two-way ANOVA with Sidak's multiple comparisons test.

Both Reviewer 1 and 3 suggested to do a supernatant pellet assay, and we have now performed differential detergent extraction of WT and *sqst-1* mutant animals expressing neuronal PolyQ aggregates that were either subjected to a hormetic heat shock or kept at control conditions. We found that a hormetic heat shock increased the soluble portion of PolyQ proteins and decreased the amount of high-molecular weight PolyQ aggregates in WT animals, but not in *sqst-1* mutants (**Supplementary Figure 5b**). This demonstrates that a hormetic heat shock has beneficial effects on neuronal proteostasis that requires *sqst-1*, similar to the conclusion reached in our FRAP experiments. Notably, WT and *sqst-1* mutant animals had similar amounts of soluble and aggregated portions of PolyQ proteins (**Figure R3**). We have now added the following on p10 of the manuscript to describe the new experiment (last sentences is listed above), and have added statistical notes in the figure legends:

“Moreover, fluorescence recovery after photobleaching (FRAP) experiments revealed that HS increased the mobility of the neuronal PolyQ aggregates in WT, but not in *sqst-1(ok2892)* mutant animals (Fig. 2d-e), while having only limited effects on diffuse neuronal PolyQ proteins (Supplementary Fig. 5a). Consistently, a hormetic HS increased the soluble portion and decreased the insoluble portion of neuronal PolyQ proteins in WT, but not in *sqst-1(ok2892)* mutant animals (Supplementary Fig. 5b). Similar to aggregate load (Fig. 2c), the baseline values for WT and *sqst-1* animals were not significantly different in FRAP (Fig. 2d-e) and differential detergent extraction experiments (Supplementary Fig. 5b).”

Figure R3: Differential Detergent Extraction of PolyQ proteins in wild-type and *sqst-1* mutants.

Fractions of total PolyQ protein detected by Western blot of neuronal PolyQ (*rgef-1p::Q40::yfp*) using a PolyQ-antibody detected in protein lysates on day 7 of adulthood in wild-type (WT) and *sqst-1(ok2892)* (*sqst-1*) animals. Protein lysates containing 200 μ g of protein were subjected to several rounds of high-speed ultracentrifugation followed by the resuspension of the pellet with increasing amounts of SDS. The last pellet resuspension was performed with 7M urea. Data shown are the mean \pm SEM from three independent experiments. $P > 0.05$, two-way ANOVA with Sidak's multiple comparisons test.

Reviewer #2 (Remarks to the Author):

My concens have been satisfactorily addressed. It is acceptable now.

Reviewer #3 (Remarks to the Author):

The authors have comprehensively addressed the critique that was raised by all referees and, in my opinion, significantly enhanced the manuscript. I therefore recommend acceptance.

Further comments from Reviewer #3 in response to Referee/Reviewer #1's evaluation:

1. The referee is unsatisfied with the answers that were provided by the authors regarding his concerns of uncontrolled expression levels of p62/SQST-1. While the authors could not detect the protein using the antibody that was provided by the Hoppe lab, they created a GFP tagged construct and followed its expression levels. Furthermore, they inhibited autophagy by Igg-1 RNAi for “several generations” in order to stabilize the protein. The referee claims that these are not physiological conditions. I agree here, but think that the over-expression of any gene is not physiological yet, this technique is commonly used to investigate the biological roles of gene products. Nevertheless, the authors can better address this critique. Since p62/SQST-1 is an unstable protein, it is not sufficient to use only 800 worms for a WB analysis (as stated in the legend of supplemental figure 6). I suggest that they will upscale the number of worms, using at

least 3000-4000 worms/lane (despite the fact that the gene is not integrated) and blot p62/SQST-1 again with the Hoppe lab's antibody. This experiment should be performed using worms that were either treated with *lgg-1* RNAi (for four generations) or left untreated.

We thank the reviewer for this experimental suggestion. As stated above (point 1), instead of scaling up the number of WT animals used in our analysis, we have now decreased the number of SQST-1-overexpressing animals and can now reproducibly compare SQST-1 levels in all strains.

2. The referee raised an additional issue asking whether the p62/SQST-1::GFP protein is functional and suggests that the authors would follow the autophagic flux. This suggestion is doable and will support the conclusions of the manuscript.

In the previous version of this manuscript, we subjected GFP-tagged SQST-1-overexpressing animals to *lgg-1/Atg8* RNAi to block autophagy, which lead to an increase in SQST-1 levels (**Supplementary Figure 7b**, formerly Supplementary Figure S6b). With the feedback from both Reviewer 1 and 3 regarding the importance of autophagy flux assays, we are no longer using this experiment to quantify SQST-1 levels, but rather discuss it as an autophagy flux assay. To this end, we have now quantified the fold increase in the levels of SQST-1 upon *lgg-1* RNAi (compared to CTRL) and included this analysis as **Supplementary Figure 7b**. We found that the SQST-1 levels were significantly increased in strains SQST-1 OE (1) and SQST-1 OE (2), whereas we did not observe an increase in SQST-1 levels in neuronal SQST-1 OE (1) strain, presumably because neurons are largely refractory to RNAi (Kamath et al, Genome Biol, 2001).

We also note that the first version of our manuscript included fluorescence images of GFP-tagged SQST-1-overexpressing animals that were subjected to autophagy gene RNAi for two generations (**Figure R3a-b**). Similarly to our Western blot analysis, these images demonstrated that autophagy gene RNAi led to an accumulation of SQST-1. Furthermore, we previously performed autophagy flux assays in which we blocked autophagy in GFP-tagged SQST-1-overexpressing animals (SQST-1 OE (1)) using the chemical autophagy inhibitor Bafilomycin A (Author Response Image 4, Chang et al. Elife, 2017) (**Figure R3c-d**). In these experiments, we found that the number of SQST-1 punctae increased upon injection of the chemical autophagy inhibitor Bafilomycin A (BafA) compared to DMSO (vehicle). Collectively, we conclude that endogenously GFP-tagged SQST-1 is turned over by autophagy, indicating that GFP-tagged SQST-1 is functional in that it can serve as a substrate for autophagy.

In conclusion, we agree with the reviewer that this is an important point, and we now emphasize it in the manuscript in line on p11 (see quote below), and we thank the reviewers for facilitating this improvement.

“To this end, we overexpressed *sqst-1* fused to GFP under the control of its endogenous promoter and found SQST-1::GFP to be expressed mainly in punctate structures in head neurons and in the pharynx throughout adulthood (**Supplementary Fig. 7a**). SQST-1::GFP accumulated upon autophagy block (**Supplementary Fig. 7b**), as observed for untagged SQST-1⁴¹, indicating that the SQST-1::GFP protein is functional as it can serve as a substrate for autophagy.”

Figure R3: SQST-1 accumulates upon autophagy block.

(a-b) GFP expression and quantification of GFP expression of animals expressing *sqst-1p::sqst-1::gfp* on day 1 of adulthood fed bacteria expressing either empty vector (CTRL) or dsRNA targeting autophagy-related genes *bec-1/BECLIN1*, *lgg-1/LC3* (RNAi (WL)) for two generations. Reduction of autophagy-related genes in SQST-1-overexpressing animals lead to an accumulation of SQST-1::GFP in the head region, as well as hypodermis, intestine and in-utero embryos, indicating that autophagy inhibition blocks the autophagic degradation of SQST-1 and leads to its accumulation. Scale bar, 200 μ m and 100 μ m for close-up of distal intestine. Expression levels relative to CTRL. Exposure time, 1.5 s for *sqst-1p::sqst-1::gfp*. N=10-12. Data are the mean \pm s.e.m. ns: $P>0.05$, **** $P<0.0001$, by one-way ANOVA with Dunnett's multiple comparisons. This experiment was repeated at least twice with similar results.

(c-d) GFP expression and quantification of GFP-positive punctae in head region of animals expressing *sqst-1p::sqst-1::gfp* on day 1 of adulthood treated with DMSO or BafA. AB = anterior bulb; TB = terminal bulb. Scale bar = 20 μ m. Data are the mean \pm s.e.m. of ≥ 30 animals combined from three experiments. * $p<0.05$ by ANOVA. Figure modified from Author Response Image 4, Chang et al. Elife, 2017.

- Finally, the referee asks the authors to better analyze the rate of polyQ aggregation. The FRAP experiment improves the analysis that was presented at the first version of the manuscript. Conducting ONE of the suggested experiments to measure the rate of aggregation, NAGE or a sup-pellet sedimentation (using a polyQ antibody) will not require much effort and better establish the manuscript.

As stated above (point 4), we have now performed Differential Detergent Extraction in WT and *sqst-1* animals expressing neuronal Q40 proteins that were either subjected to a hormetic heat

shock or left untreated. Using this new assay, we were able to demonstrate that the hormetic heat shock increases the soluble portion of PolyQ aggregates in WT animals, but not in *sqst-1* mutant animals. This new analysis demonstrates that a hormetic heat shock has beneficial effects on neuronal proteostasis that requires *sqst-1*, we have included this analysis as **Supplementary Figure 5b**.

REVIEWERS' COMMENTS:

Reviewer #1 (Remarks to the Author):

The authors certainly made an effort to address the concerns. I have still doubts though. The data obtained are not robust - e.g. just concerning now their added data set on the differential detergent extraction (which only means that they used different amounts of SDS or once 7 M urea) to resolubilise polyQ (Sup 5b).

1. A control (untreated) is missing.
2. A control for the antibody is missing - e.g. just loading a YFP-expressing lysate to test which bands are specific for the antibody.
3. I don't know why they used a gradient of SDS when they only detect signals in 0.1% and from then on less and less? This means that probably all SDS-sensitive polyQ moiety has already been released upon 0.1%. And last they used 7M urea.

What do the authors think the HMW species is? It migrates around the 100 MW marker suggesting it is a trimer of polyQ. I would have expected to observe signals in the pockets of the SDS gel (and then Western blot). Again a control without treating the samples with SDS or Urea would have helped here.

With the control worms they observed significant differences for only 2 conditions: 0.1% SDS and 7M urea - why did they not analyse more detergents or time points of treatment etc (Triton X-100, formic acid etc).

And also sqst-1 shows for those 2 conditions a difference - just not significant (enough)?

The Western blot above does not really fit to the quantification. The blot for the control is under-exposed (compare the overall signal intensities).

This data set is a good example that for most of the presented data - you'll detect some flaws.

Reviewer #3 (Remarks to the Author):

I think that the key issues of measuring polyQ40 aggregation and controlling the expression levels of SQST-1::GFP have been satisfactorily addressed. The differential solubility assay (Figure S5B of the revised manuscript) establishes the claim that a short heat shock modulates the aggregation of neuronal polyQ40. The new analysis of SQST-1 and SQST-1::GFP levels (Figure S7, b-d of the revised manuscript) is acceptable. In conclusion, I think that the manuscript should be accepted for publication in its current form.

Ehud Cohen

Response to Reviewer 1:

Reviewer 1 – Comments to the author:

The authors certainly made an effort to address the concerns. I have still doubts though. The data obtained are not robust - e.g. just concerning now their added data set on the differential detergent extraction (which only means that they used different amounts of SDS or once 7 M urea) to resolubilise polyQ (Sup 5b).

We respectfully disagree with the Reviewer that the new differential detergent data are not robust and would like to reiterate how the differential detergent extraction was performed, since it appears from the Reviewer's questions that the nature of our assay may have been misunderstood. To aid in this, we provide a schematic outline of the experiment in **Figure R1** (below). The Differential Extraction Assay of *C. elegans* lysates represents an expanded and more informative version of a supernatant pellet assay, and was carried out with three independent biological samples per condition.

In brief, in this sequential supernatant and pelleting assay, the SDS concentration was incrementally increased in the buffer used to resolubilize the pellet in each step. To finally solubilize PolyQ aggregates that are insoluble in the highest concentration of 2% SDS, a very strong detergent buffer containing 2% SDS, 1% Triton X-100, 7M urea, and 2M thiourea was used to fully solubilize any remaining PolyQ aggregates. Since we performed these extractions on the same *C. elegans* lysates in sequential steps, not on individual lysates, this assay is a readout for the amount of aggregates that are soluble in detergents of different strength, hence we can make direct and meaningful comparisons of the extent of solubility and therefore aggregation status of PolyQ proteins in our experimental conditions. Importantly, we found that PolyQ proteins form high-molecular weight (HMW) species, as previously reported (Nollen et al, PNAS, 2004; Imanikia et al, Curr Biol. 2019) that are only present in the last resuspension step using urea buffer, suggesting that they are insoluble in milder detergents. Since we find that a hormetic heat shock increases the soluble portion of PolyQ aggregates (in 0.1% SDS buffer) and decreases the HMW species of PolyQ proteins (in urea buffer) in WT animals but not in *sqst-1* mutants, this assay further supports our conclusion that a hormetic heat shock improves proteostasis in animals expressing neuronal PolyQ40 proteins in an *sqst-1*-dependent manner.

1. A control (untreated) is missing.

We do not share this concern because this experiment aimed at comparing whether a hormetic heat shock early in life would have effects on the soluble and insoluble fraction of PolyQ proteins in WT animals, and, if so, investigate if any such heat-shock mediated differences would be observed in *sqst-1* mutants. We designed the experiment based on findings by multiple labs that aggregated PolyQ proteins are insoluble in buffers containing up to 2% SDS using Filter trap experiments (Koyuncu S. et al. Nat Commun, 2018, Vilchez et al, Nature 2012, Sin et al, Mol Cell, 2017; Sin O et al., Bio Protoc. 2018), and supernatant pellet assay (Nollen et al, PNAS, 2004). Furthermore, previously performed differential detergent extraction assays (Moronetti et al, PNAS, 2012) were designed similarly, determining the solubility of PolyQ proteins in an SDS gradient ranging from 0.1%-2% SDS. Since we find that the HMW species of PolyQ proteins are only present in our last resuspension step using urea buffer, we believe that we can make meaningful comparisons, including with the 0.1% SDS buffer as the lowest concentrated protein extraction buffer.

Figure R1. Schematic of Differential Detergent Extraction procedure. 1-day old WT and *sqst-1* animals expressing neuronal Q40 proteins (*rgef-1p::Q40::yfp*) were either subjected to a 1 h heat shock at 36°C, or kept at control conditions. Animals were aged until day 7 of adulthood, on which 100 µl of worm pellet (~30,000 animals) was mixed with 100 µl of 0.5 mM Zirconia beads and 200 µl of lysis buffer (10 mM Tris-HCl, pH 8.0, 150 mM NaCl, 0.1% SDS, Complete Proteinase Inhibitor with 2 mM EDTA, 0.5 mM PMSF) was added. Lysates were generated by homogenization using a Fastprep homogenizer (3 x 6000 rpm, 45 s, with 30 s pause). 200 µg of protein lysate was subjected to ultracentrifugation (100,000 g, 20 min, 4°C). The supernatant was transferred to a fresh tube and kept for Western blot analysis. The pellet was resuspended in 100 µl lysis buffer with increasing SDS concentration (0.25%, 0.5%, 0.75%, 1.0%, or 2% SDS), followed by another centrifugation step (100,000 g, 20 min, 4°C). After the last centrifugation step, the pellet was dissolved in 100 µl urea buffer (30 mM Tris-HCl, pH 8.0, 150 mM NaCl, 2% SDS, 1% Triton X-100, 7 M urea, 2 M thiourea, and 0.5 mM PMSF, Complete Proteinase Inhibitor). The samples were loaded in corresponding order on a SDS-gel (Lane 1-7, as indicated in the figure).

2. A control for the antibody is missing - e.g. just loading a YFP-expressing lysate to test which bands are specific for the antibody.

We have carried out a control experiment for the PolyQ antibody that we used in this study. Specifically, we performed Western blot analyses of *C. elegans* expressing either neuronal PolyQ::YFP proteins (either a stretch of Q40 or Q67), muscle Q35::YFP proteins, or animals not expressing any transgenic PolyQ proteins, such as WT and long-lived *daf-2* insulin/IGF-1 mutants (noting these controls were conducted as controls for other experiments not related to this project) on either day 1 or day 5 of adulthood (**Figure R2**). The HMW species were observed in transgenic animals expressing PolyQ proteins, while no signal was detected in WT or *daf-2* animals. This observation is consistent with the PolyQ antibody specifically detecting PolyQ proteins. We note that the observed HMW band seems to increase with age, which is consistent with an age-dependent increase in PolyQ aggregates in these animals (Brignull HR et al. Methods Enzymol. 2006). Since we used a PolyQ-specific antibody for this assay instead of a GFP- or YFP- specific antibody, we believe that WT animals are an appropriate control, rather than animals expressing YFP.

Figure R2: PolyQ antibody detects PolyQ species only in transgenic animals expressing PolyQ proteins. Western blot of WT animals on day 1 and day 5 of adulthood expressing neuronal PolyQ (*rgef-1p::Q40::yfp* (Q40), *rgef-1p::Q67::yfp* (Q67)) (left panel), WT and *daf-2* mutants on day 1 and day 5 of adulthood expressing muscle PolyQ (*unc-54p::Q35::yfp* (Muscle Q35)), and WT and *daf-2* mutants on day 1 and day 5 of adulthood (without any transgene) (right panel). These Western blots were performed in the same experiment and 50 animals were used per sample and antibodies against PolyQ and tubulin were used.

3. I don't know why they used a gradient of SDS when they only detect signals in 0.1% and from then on less and less? This means that probably all SDS-sensitive polyQ moiety has already been released upon 0.1%. And last they used 7M urea.

When we performed filter trap experiments (according to protocol Sin O et al., Bio Protoc. 2018) on WT and *sqst-1* animals expressing neuronal PolyQ40, we only detected a very limited amount of aggregates that were insoluble in 2% SDS (Data presented in Rebuttal 1). Because of these results, we decided to perform a differential detergent extraction using buffers including a range of SDS concentrations below 2% SDS. As mentioned above, we designed our experiment based on previously performed differential detergent extraction assays (Moronetti et al, PNAS, 2012) that used an SDS gradient ranging from 0.1%-2% SDS to determine the solubility of PolyQ proteins in *C. elegans*. Since we find that a hormetic heat shock increases the soluble portion of PolyQ aggregates (in 0.1% SDS buffer) and decreases the HMW species of PolyQ aggregates (in urea buffer) in WT animals, but not in *sqst-1* mutants, this assay supports our conclusion that a hormetic heat shock improves proteostasis in animals expressing neuronal Q40 proteins in an *sqst-1*-dependent manner.

4. What do the authors think the HMW species is? It migrates around the 100 MW marker suggesting it is a trimer of polyQ.

PolyQ proteins have been previously shown to form HMW forms, and our data are consistent with these observations (Nollen et al, PNAS, 2004; Imanikia et al, Curr Biol. 2019). While we are also very interested in the nature of these HMW species of PolyQ proteins (which could be either aggregated forms of PolyQ proteins

that are not fully denatured even in the strongest detergent we used, or caused by post-translational modifications that change the mobility of the PolyQ proteins), we believe that such a characterization is beyond the scope of this manuscript.

I would have expected to observe signals in the pockets of the SDS gel (and then Western blot). Again a control without treating the samples with SDS or Urea would have helped here.

Since the lysates were separated into pellets and supernatant at every step (**Figure R1**), only the soluble material in the supernatants was loaded onto the gel and no insoluble proteins will be present in these samples. The last resuspension step was performed with a very strong detergent buffer containing 2% SDS, 1% Triton X-100, 7M urea, and 2M thiourea, designed to solubilize any remaining aggregated PolyQ proteins.

5. With the control worms they observed significant differences for only 2 conditions: 0.1% SDS and 7M urea - why did they not analyse more detergents or time points of treatment etc (Triton X-100, formic acid etc).

As mentioned above, aggregated PolyQ proteins have been found to be insoluble in buffers containing 1% SDS, and since we were interested in separating the soluble and insoluble fraction of PolyQ proteins and compare them in our experimental conditions, we designed our experiments to include concentrations below and above 1% SDS, with the last step designed to solubilize aggregates species using a very strong detergent buffer containing 2% SDS, 1% Triton X-100, 7M urea, and 2M thiourea. We will update the figure legend to include the full composition of the urea buffer that was used in the last resuspensions step to avoid any confusion. We can also add the schematic of the extraction scheme (**Figure R1**) into the Figure to help clarify the assay we performed.

6. And also *sqst-1* shows for those 2 conditions a difference - just not significant (enough)?

We conducted three experiments with independently collected lysates of 7-day old WT animals and *sqst-1* mutants that were either untreated or subjected to a hormetic heat shock. We show the blots from one experiment and the quantification of all three data sets. Our quantification showed that there was no statistically significant difference between WT-CTRL and *sqst-1*-CTRL in any fraction (as also shown in Figure R3 in Rebuttal 2). Furthermore, there was no statistically significant difference in any fraction between *sqst-1*-CTRL and *sqst-1*-HS. The only differences that we found were in the comparison of WT-CTRL and WT-HS using 0.1% SDS extraction buffer and in the high molecular weight species present using urea buffer as an extraction buffer. Below, we include a summary of all P-values in **Table R1**.

These data suggest that hormetic heat shock increases the soluble portion of PolyQ aggregates and decreases the HMW species of PolyQ aggregates in WT animals, but not in *sqst-1* mutants.

Table R1: Summary of P-values comparing the mean PolyQ fractions using different extraction buffers of three independent experiments using two-way ANOVA with Sidak's multiple comparisons test.

Extraction Buffer	P-value		P-value WT-CTRL vs. WT-HS
	WT-CTRL vs. sqst-1 -CTRL	sqst-1 -CTRL vs. sqst-1 -HS	
0.1% SDS	0.9997	0.9982	0.0085 (**)
0.25% SDS	0.8418	>0.9999	>0.9999
0.5% SDS	>0.9999	0.9848	>0.9999
0.75% SDS	>0.9999	>0.9999	>0.9999
1% SDS	>0.9999	>0.9999	>0.9999
2% SDS	>0.9999	>0.9999	>0.9999
Urea buffer (2% SDS, 7M Urea) – low molecular weight	>0.9999	>0.9999	>0.9999
Urea buffer (2% SDS, 7M Urea) ¹ – high molecular weight	0.7327	0.5807	0.0137 (*)

¹ Urea buffer contained 2% SDS, 1% Triton X-100, 7 M urea, and 2 M thiourea (*, indicates significant P values).

7. The Western blot above does not really fit to the quantification. The blot for the control is under-exposed (compare the overall signal intensities).

We used the same amount of total protein for the differential detergent extraction experiments and all Western blots were exposed the same amount of time. We would like to clarify how the quantification of the PolyQ signals in each fraction was performed. Noting again that the same lysate is subjected to sequential treatments in this protocol, the quantification of protein present in each detergent condition can be calculated as the fraction of total protein (equals total signal of all detergent concentrations) and is thus independent of exposure time. Consequently, comparisons of soluble and insoluble PolyQ species as a fraction of total PolyQ proteins can be made.

This data set is a good example that for most of the presented data - you'll detect some flaws.

We disagree with the reviewer that any of the presented data in our manuscript have flaws. Each experiment has been designed, executed, and analyzed while observing rigor and reproducibility so that statistically meaningful conclusions can be made from the obtained data. Subsequently, we have aimed to appropriately discuss any possible limitations of our conclusions, inherently associated with experimental investigations of a living organism.

Reviewer 2 – Comments to the author:

I think that the key issues of measuring polyQ40 aggregation and controlling the expression levels of SQST-1::GFP have been satisfactorily addressed. The differential solubility assay (Figure S5B of the revised manuscript) establishes the claim that a short heat shock modulates the aggregation of neuronal polyQ40. The new analysis of SQST-1 and SQST-1::GFP levels (Figure S7, b-d of the revised manuscript) is acceptable. In conclusion, I think that the manuscript should be accepted for publication in its current form.

Ehud Cohen

We thank the reviewer for their evaluation of the newly added experiments.